# Systematic dissection of tumor-normal single-cell ecosystems across a thousand tumors of 30 cancer types

Junho Kang[1,9], Jun Hyeong Lee[2,9], Hongui Cha[3], Jinhyeon An [2], Joonha Kwon [2,4], Seongwoo Lee[1], Seongryong Kim[1], Mert Yakup Baykan [1], So Yeon Kim[2], Dohyeon An[2], Ah-Young Kwon [5], Hee Jung An[5], Se-Hoon Lee [3,6] ✉, Jung Kyoon Choi [2,7] ✉ & Jong-Eun Park [1,8] ✉

The complexity of the tumor microenvironment poses significant challenges in cancer therapy. Here, to comprehensively investigate the tumor-normal ecosystems, we perform an integrative analysis of 4.9 million single-cell transcriptomes from 1070 tumor and 493 normal samples in combination with pan-cancer 137 spatial transcriptomics, 8887 TCGA, and 1261 checkpoint inhibitor-treated bulk tumors. We define a myriad of cell states constituting the tumor-normal ecosystems and also identify hallmark gene signatures across different cell types and organs. Our atlas characterizes distinctions between inflammatory fibroblasts marked by *AKR1C1* or *WNT5A* in terms of cellular interactions and spatial co-localization patterns. Co-occurrence analysis reveals interferon-enriched community states including tertiary lymphoid structure (TLS) components, which exhibit differential rewiring between tumor, adjacent normal, and healthy normal tissues. The favorable response of interferon-enriched community states to immunotherapy is validated using immunotherapy-treated cancers ($n = 1261$) including our lung cancer cohort ($n = 497$). Deconvolution of spatial transcriptomes discriminates TLS-enriched from non-enriched cell types among immunotherapy-favorable components. Our systematic dissection of tumor-normal ecosystems provides a deeper understanding of inter- and intra-tumoral heterogeneity.

Tumors are highly heterogeneous entities composed of malignant cells and diverse tissue-infiltrating stromal and immune cells that form the tumor microenvironment (TME)[1]. The advent of single-cell RNA sequencing (scRNA-seq) technologies has provided unbiased and systematic molecular profiling for high-resolution characterization of extensive heterogeneity embedded in the TME[2–4].

Molecular and cellular heterogeneity within the TME collectively influences various aspects of tumors, including progression, metastasis,

[1]Graduate School of Medical Science and Engineering, Korea Advanced Institute of Science and Technology, Daejeon, Republic of Korea. [2]Department of Bio and Brain Engineering, Korea Advanced Institute of Science and Technology, Daejeon, Republic of Korea. [3]Division of Hematology-Oncology, Department of Medicine, Samsung Medical Center, Sungkyunkwan University School of Medicine, Seoul, Republic of Korea. [4]Division of Cancer Data Science, National Cancer Center, Bioinformatics Branch, Goyang, Republic of Korea. [5]Department of Pathology, CHA Bundang Medical Center, CHA University, Seongnam-si, Republic of Korea. [6]Department of Health Sciences and Technology, Samsung Advanced Institute of Health Science and Technology, Sungkyunkwan University School of Medicine, Seoul, Republic of Korea. [7]Penta Medix Co., Ltd., Seongnam-si, Gyeonggi-do, Republic of Korea. [8]Biomedical Research Center, Korea Advanced Institute of Science and Technology, Daejeon, Republic of Korea. [9]These authors contributed equally: Junho Kang, Jun Hyeong Lee. ✉e-mail: shlee119@skku.edu; jungkyoon@kaist.ac.kr; jp24@kaist.ac.kr

and treatment response[1]. As evidence of intra-tumoral and inter-tumoral heterogeneity mounts, various attempts are being made to compile consensus gene signatures at the pan-cancer level. For example, a pan-cancer atlas of T, myeloid, and malignant cells has recently been published[5–8]. Although these pan-cancer analyses well characterize the cell types of interest, complex interactions among the TME components and the distinctions from paired normal tissues have not yet been fully appreciated, leading to a limited perspective of tumor heterogeneity, neglecting or oversimplifying potentially crucial molecular and cellular interactions. Indeed, TME phenotypes are not simply binarized into anti-tumor or pro-tumor but rather represent interactive cellular organizations or ecosystems[9]. Targeting tumor-specific interactions underlying tumor ecosystems presents an appealing strategy that can yield synergistic therapeutic effects[9,10]. Therefore, it is crucial to dissect the intricate and multilayered ecosystems across diverse cancer and tissue types to develop more efficient therapeutic strategies.

Cancer immunotherapy based on checkpoint blockade has emerged as a promising therapeutic strategy with a profound impact on cancer treatment. However, the heterogeneous nature of tumor ecosystems poses one of the main remaining challenges, that is, the varying efficacy of checkpoint inhibitors across cancer types and patients[11]. Recent studies have highlighted interferon signatures and tertiary lymphoid structure (TLS), an ectopic aggregate of immune cells with a lymphoid-like structure, as a key determinant of responses to immunotherapy[12–14]. Nevertheless, a detailed understanding of components favorable for immunotherapy that are associated with TLS remains elusive. Therefore, a comprehensive pan-cancer dissection of tumor ecosystems is necessary to discriminate TLS-enriched and non-enriched cell types that confer favorable responses to immunotherapy, elucidate the mechanisms by which these components shape tumor immunity, and unravel the interactions among them. Such insights will enable us to better comprehend the differential response to immunotherapy observed in patients of diverse cancer types.

Herein, we construct an unsorted tumor-normal single-cell transcriptomic atlas covering 30 cancer types and 4.9 million cells from 1070 tumors and 493 normal samples. Our analysis incorporates diverse analytical approaches including the AND-gating algorithm and non-negative matrix factorization (NMF) visualization at single-cell resolution to unveil the distinctions between tumor/normal ecosystems. We outline hallmark gene signatures across diverse cell types and organs. Our analysis reveals the heterogeneity of inflammatory fibroblasts, including *CXCL1/3/8* expressing *AKR1C1*+ and *WNT5A*+ inflammatory fibroblasts, which exhibit distinct organ allocations, tissue preferences, cellular interactions, and spatial co-localization patterns. By analyzing the co-occurrence patterns of cell states, we have uncovered tumor-specific rewiring of interferon-enriched community which comprise TLS components including *CCL19*+ fibroblast and *LAMP3*+ DC that hold distinct clinical significance in immunotherapy-treated cohorts ($n = 1261$), including our lung cancer (LC) cohort ($n = 497$). Furthermore, we categorize cell types enriched in TLS and those that are not within the spectrum of immunotherapy-favorable components using spatial transcriptome analysis and derive a TLS signature that predicts favorable responses to immunotherapy. In summary, our pan-cancer meta-atlas provide deeper insights into tumor-normal ecosystems and serve as a valuable resource for the development of diagnostic and therapeutic strategies. We have deposited the comprehensively analyzed datasets to the Zenodo repository (DOI:10.5281/zenodo.10651059) and our atlas can be interactively visualized at https://cellatlas.kaist.ac.kr/ecosystem/.

## Results
### Construction of a pan-cancer tumor-normal single-cell meta-atlas
To generate a comprehensive census of the tumor and normal ecosystems, we selected published scRNA-seq datasets on cancer,

adjacent normal, and healthy normal samples that have not been enriched for specific cell types. As a result, a tumor-normal single-cell transcriptomic atlas encompassing 30 different cancer types across 104 datasets was constructed (Supplementary Data 1). After data curation, this meta-atlas covered 4.9 million cells from 1,070 tumors and 493 normal samples derived from 999 donors (Fig. 1A). Breast cancer (BRCA) was the most abundant cancer type, followed by LC, head and neck cancer (HNSC), and hepatocellular carcinoma (HCC).

To profile a variety of ecosystems across diverse cancer and tissue types, entire collected single-cell datasets were integrated and annotated at a global scale. Then, an AND-gating algorithm (see Methods) was applied to characterize differentially expressed genes that are cell type-specific and universally found in tumor and normal tissues of various organs. Subsequently, the annotated gene expression matrices were then split into major cell types and decomposed into various cell states using an NMF analysis[3,8] (Fig. 1B–D and Supplementary Fig. S1A, B). To maximize the recovery of rare cell states, NMF modules were collected per individual samples by scanning multiple parameters, and the resulting modules were clustered and projected to Uniform Manifold Approximation Projection (UMAP) to search recurring consensus modules.

Among the clusters of NMF modules (cell states), we identified the ones representing the contaminations from ambient RNAs or doublets and removed those clusters from further analysis using an automated pipeline (Fig. 1E and Supplementary Fig. S1C). NMF module clusters enriched with ribosomal/mitochondrial genes were also filtered out. The final UMAP representation of NMF modules demonstrates the overall structure of recurring cell states across multiple samples (Fig. 1E). Utilizing UMAP representation for NMF modules allows us to visually inspect the characteristics of each cell state by their origin (e.g., tissue type, organ, etc.). The cell states were defined and annotated based on genes with the highest average NMF weights (Supplementary Data 2). This allowed us to dissect tumor single-cell and bulk transcriptome data using the cell state filters derived at single-cell resolution (Fig. 1E, Cell state heterogeneity) and monitor their co-occurrence across normal and tumor samples to identify potential interactions between cell types and their contribution to the TME (Fig. 1E, Coincidence). We also applied the cell state signatures to deconvolute bulk RNA-seq cohorts (TCGA, immune checkpoint therapy) to check their clinical implications (Fig. 1E, Survival analysis). Finally, we projected cells using our cell states as a reference component to assess the correspondence between cell states and cell types[15] (Fig. 1E, Cell projection to reference components) and deconvoluted spatial transcriptomics across 11 cancer types ($n = 137$) using our cell types (Fig. 1E and Supplementary Data 3, Spatial transcriptomics).

### Identification of universal hallmark gene signatures of tumor-normal ecosystems
An AND-gating algorithm (see Methods) was implemented to systematically characterize hallmark genes that are recurrently upregulated in tumors compared with normal tissues, and vice versa, in major cell types comprising the TME of diverse organs (Supplementary Fig. S2A and Supplementary Data 4). For CD8+ T cells, co-stimulatory molecule (*CD27*) and immune checkpoint or exhaustion markers such as *CXCL13*, *PDCD1*, *TIGIT*, *CTLA4*, *LAG3*, and *TNFRSF9* were commonly elevated in tumors whereas *IL7R*, *PTGER2*, and *PTGER4* were elevated in normal tissues (Fig. 2A, B). Of note, CD8+ T cells of pancreatic tumor tissues did not show upregulation of *PDCD1* and *LAG3*, which potentially accounts for the current inapplicability of immune checkpoint inhibitors in pancreatic cancer (PAAD) in contrast to other cancer types[16]. Similarly, tumor-associated NK cells were marked by *ZNF683* and *KRT81*. Tregs in tumors elevated genes with regulatory functions such as *RBPJ*, *CXCR3*, and *ZBED2* whereas Tregs in normal tissues upregulated *CCR7* and *CXCR5*, indicating distinct mechanisms for immune cell

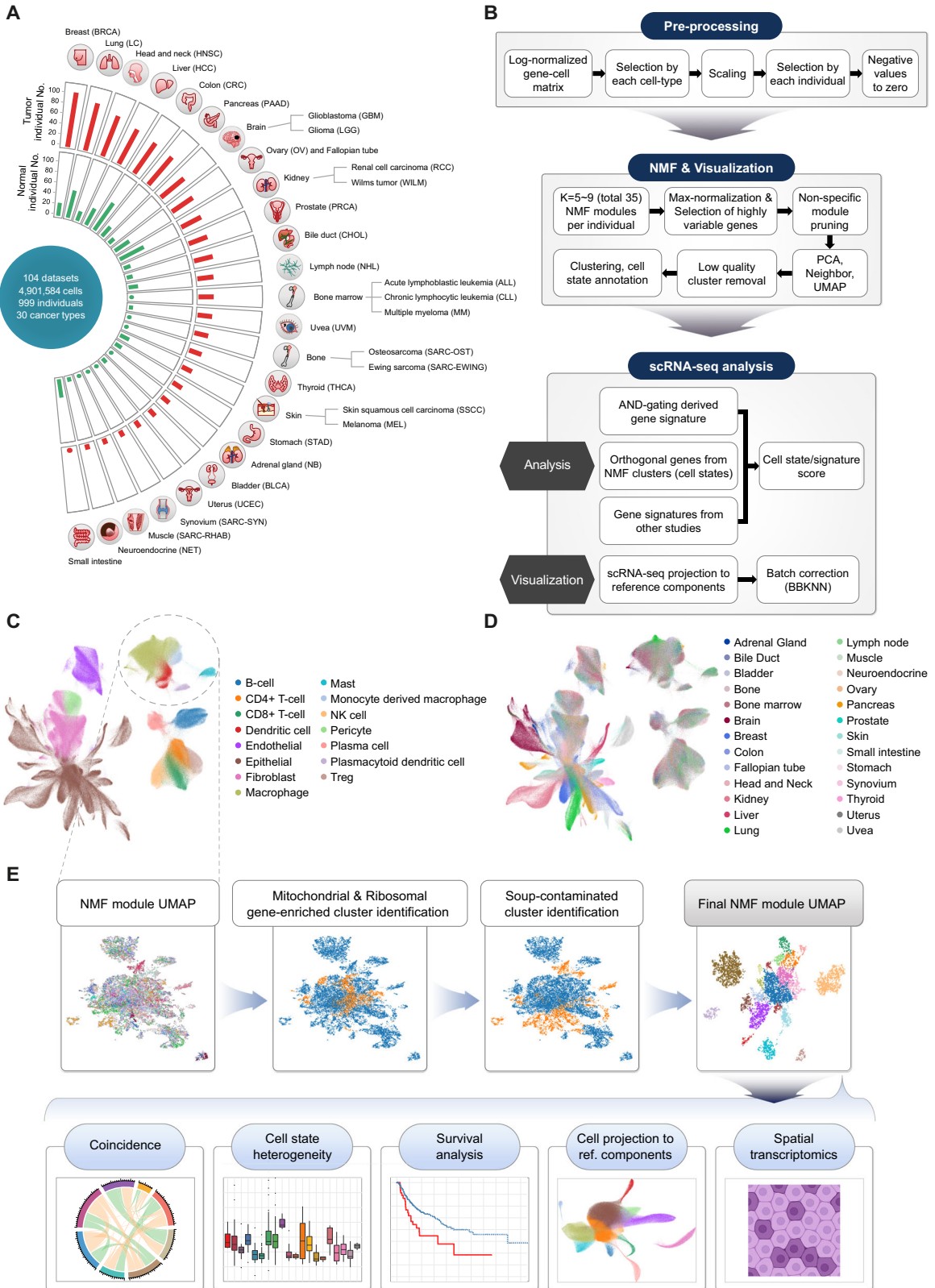

**Fig. 1 | Overview of the pan-cancer tumor-normal landscape at single-cell resolution. A** Overview of the scRNA-seq cohort across the 30 cancer types collected in this study. **B** Workflow of the tumor-normal single-cell transcriptomic atlas and NMF processing. UMAP visualization of a subset of the tumor-normal single-cell atlas colored by (**C**) cell types and (**D**) organ origins. **E** Graphical clustering schematics of NMF modules with automatized soup-effect detection algorithm and subsequent analyses.

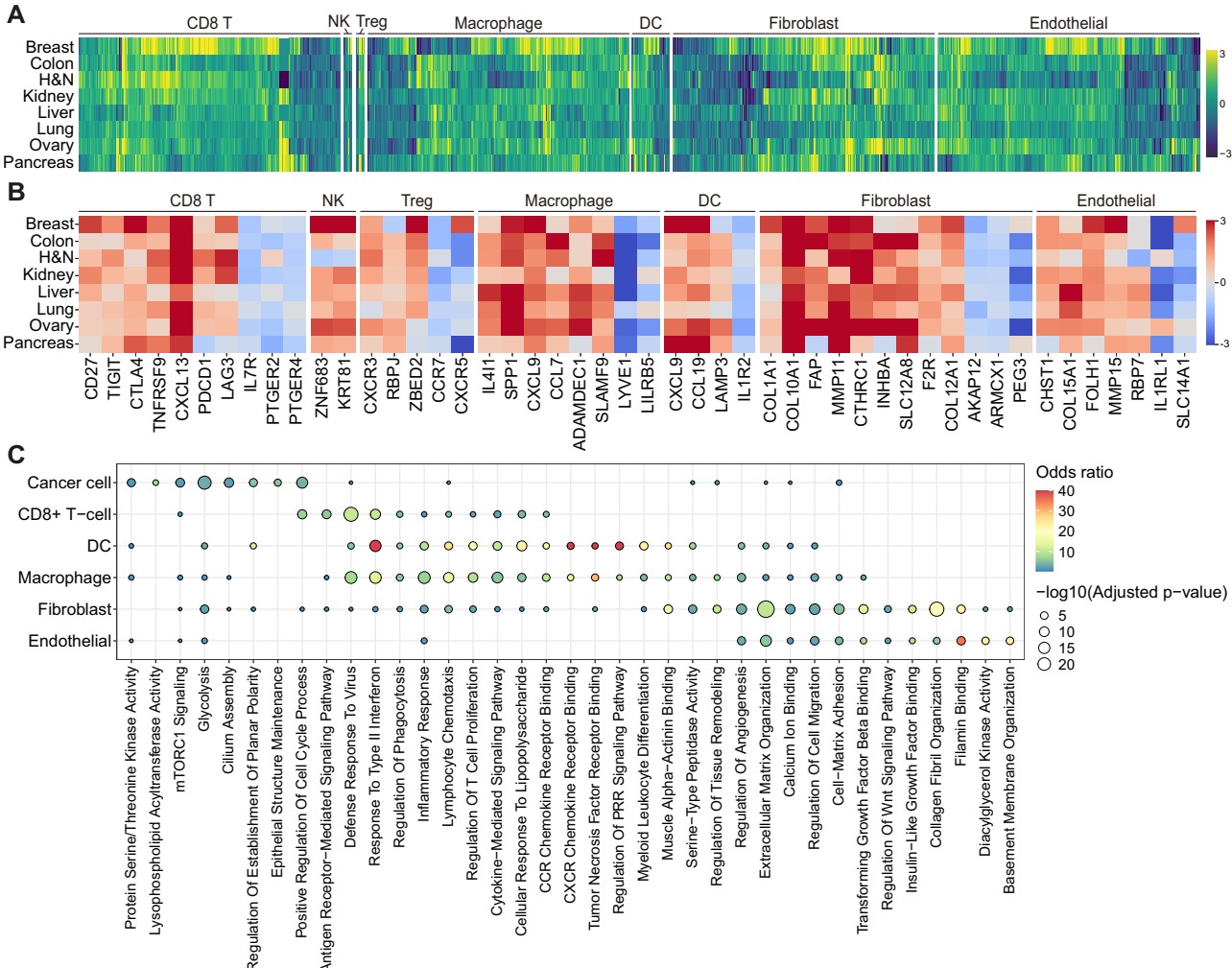

**Fig. 2 | Landscape of hallmark genes of tissue ecosystems across organs.**
**A** Characterization of hallmark genes in tumor and normal ecosystems across organs. Cell type is noted on top of the heatmap and only the genes that are upregulated in four or more cancer types are depicted. Each box in the heatmap represents log$_2$ fold-change values with positive values indicating upregulation in tumors. H&N, head and neck; DC, dendritic cell; NK, natural killer cell; Treg, regulatory T cell. **B** Detailed heatmap of hallmark genes identified in Fig. 2A across cancer and cell types. Each color in the heatmap represents log$_2$ fold-change values with positive values indicating upregulation in tumors. H&N, head and neck; DC, dendritic cell; NK, natural killer cell; Treg, regulatory T cell. **C** Gene ontology analysis of tumor hallmark genes of diverse cell types using Enrichr. The color and size of the dot represent the odds ratio and *p* value from two-sided Fisher's exact test, adjusted with the Benjamini-Hochberg method, respectively. PRR, pattern recognition receptor.

recruitment and infiltration[17] (Fig. 2A, B). In particular, tumor-infiltrated macrophages universally expressed immune checkpoint (*IL4I1*)[18], M2 polarization-related (*SPP1*)[5,19], and inflammatory genes (*CCL7*, *ADAMDEC1*, and *SLAMF9*), whereas dendritic cells in tumors were found to exhibit elevated expression of *CCL19* and *LAMP3*, which are associated with inflammatory and migratory functions (Fig. 2A, B). Gene ontology (GO) analysis revealed that genes upregulated in tumor-infiltrating macrophages, dendritic cells, and CD8+ T cells were enriched in related functions and pathways including defense responses to viruses, response to type II interferon, inflammatory responses, chemotaxis of lymphocytes, and cytokine-mediated signaling pathways (Fig. 2C).

As for non-immune cell types, cancer cells universally manifested protein serine/threonine kinase activity (*PRKCA*, *GSK3B*, and *CAMKK2*), glycolysis (*PLOD1*, *EGLN3*, and *P4HA1*), mTORC1 signaling (*SLC2A1*, *GMPS*, and *PDK1*), and GO terms associated with the positive regulation of the cell cycle process (*E2F7*, *E2F8*, and *KIF23*; Fig. 2C and Supplementary Fig. S2B). Cancer-associated fibroblasts (CAF) expressed well-known markers including *FAP*, *COL1A1*, *COL10A1*, *MMP11*, and *CTHRC1*[20,21], and other genes such as *INHBA*, *SLC12A8*, *F2R*, and

*COL12A1* in various organs (Fig. 2A, B). Tumor endothelial cells upregulated angiogenesis-associated genes including *CHST1*, *FOLH1*, and *MMP15*[22–24]. Both tumor-associated fibroblasts and endothelial cells were enriched in terms related to extracellular matrix organization, regulation of cell migration, cell-matrix adhesion, and filamin binding (Fig. 2C). Overall, these results outline dysregulated hallmark signatures of the TME components.

## Deconvolution of tumor-normal ecosystems into heterogeneous cell states

The systematic dissection of the complex tumor and normal ecosystems identified a myriad of cell states or co-regulated genes that are strongly consistent with previously reported signatures and those that have not been yet identified in the previous pan-cancer analyses[5] (Fig. 3A and Supplementary Fig. S3, 4). For myeloid cell states, we noted *CTSK*+ macrophage (*SLC9B2* and *CTSK*), *CXCL9*+ macrophage (*CXCL9* and *ENPP2*), Langerhans cell (*CD1A* and *CD2O7*), mononuclear phagocyte (*DLEU2* and *FMN1*), and PRR-induced activation state marked by chemokines (*CXCL1* and *CXCL5*), migratory (*SLAMF1*) and immunoregulatory markers (*ITGB8*). *SLAMF1* and *ITGB8* are

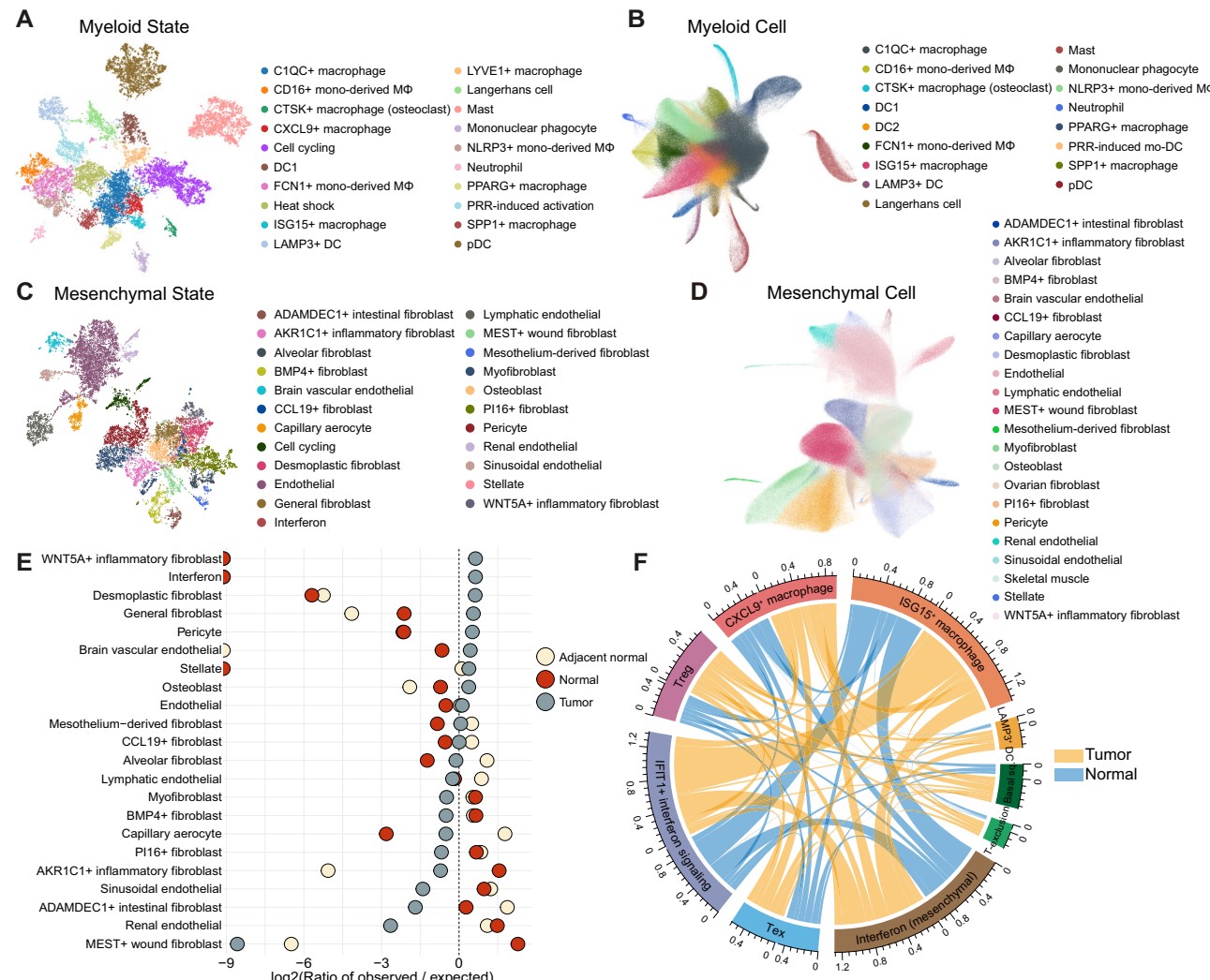

**Fig. 3 | Deconvolution of the tumor-normal ecosystem into heterogeneous cell states.** UMAP visualization of (**A**) myeloid cell states and (**B**) corresponding reference component analysis of myeloid cells. NMF modules were graphically clustered and colored according to cell states (**A**). Then, cells were mapped to the reference components composed of myeloid cell state genes (**B**). DC, dendritic cell; mono-derived MΦ, monocyte-derived macrophage; mo-DC, monocyte-derived dendritic cell; pDC, plasmacytoid dendritic cell; PRR, pattern recognition receptor. UMAP visualization of (**C**) mesenchymal cell states and (**D**) corresponding reference component analysis of mesenchymal cells. NMF modules were graphically clustered and colored according to cell states (**C**). Then, cells were mapped to the

reference components composed of mesenchymal cell state genes (**D**). **E** Ro/e analysis of tissue enrichments in mesenchymal cell states. The dotted vertical line represents where Ro/e is zero. **F** Circos plot illustrating co-occurrences between the cell states in normal (blue) and tumor (yellow) tissues. The length of the arcs represents the sum of co-occurrences with other cell states in adjacency. A longer arc indicates more frequent co-occurrence with other cell states. Basal sq, basal squamous state. DC, dendritic cell; Tex, exhausted CD8[+] T cell; T-exclusion, T cell exclusion program; Treg, regulatory T cell. Source data are provided as a Source Data file.

upregulated during TLR-induced DC maturation[25–28], thus named PRR-induced activation. We also identified various B cell states, including the germinal center B cell (GCB; *SUGCT* and *RGS13*) and plasma cell precursor (*FNDC3B* and *FNDC3A*) states (Supplementary Fig. S3F). Evaluation of tissue enrichment of immune cell states with Ro/e analysis revealed preferences of exhausted CD8[+] T cell (*LAG3* and *TOX*), *SPP1*[+] (*SPP1* and *ANGPTL4*), *CTSK*[+], and *CXCL9*[+] macrophage states in tumors (Supplementary Fig. S5A, B)[5]. GCB and plasma cell states were abundant in adjacent normal tissues, while PRR-induced activation states were enriched in healthy normal tissues (Supplementary Fig. S5B, C). Utilizing cell type specific cell state profiles as a reference for embedding, we identified cell types that well reflect corresponding cell states (Fig. 3B and Supplementary Fig. S6, 7): the PRR-induced activation state was captured as a PRR-induced mo-DC cell cluster that predominantly originated from gynecological cancers such as ovarian cancer (OV) and uterine corpus endometrial carcinoma (UCEC)

patients, consistent with the distribution of PRR-induced activation state score (Supplementary Fig. S8).

The epithelial and neural cells were deconvoluted into 35 states (Supplementary Fig. S9) and the subsequent reference component analysis using these cell states clustered cells by their origins (Supplementary Fig. S10A, B). Epithelial hallmark programs such as cycling (*TOP2A* and *BIRC5*) and stress (*PPP1R15A* and *KLF5*) were found. Regarding previously cataloged signatures[7,29], renal cell carcinoma (RCC) and glioblastoma manifested high levels of hypoxia and metal response while low-grade glioma, UCEC, and neuroblastoma exhibited low levels of partial epithelial-to-mesenchymal transition (pEMT), oxidative phosphorylation, and antigen-presenting machinery signatures, respectively (Supplementary Fig. S10C, D).

Notably, we classified and annotated diverse cell states of mesenchymal origin, including the *CCL19*[+] fibroblast (*CCL19* and *CXCL13*), *PI16*[+] fibroblast (*MFAP5* and *IGFBP6*), myofibroblast (*ACTA2*

and *MYH11*), desmoplastic fibroblast (*LRRC15* and *MMP11*), and fibroblast states exclusive to specific organs that potentially reflect organ-specific functions (Fig. 3C, D and Supplementary Fig. S11). These states were also identified as a variety of mesenchymal subpopulations when projected onto the reference components. While the desmoplastic fibroblast state was highly tumor-specific, the myofibroblast populations were evenly distributed across non-malignant tissues, in contrast to the previous reports suggesting myofibroblasts as major CAFs (Fig. 3E)[30,31].

These diverse cell states are collectively engaged to shape the complex tumor ecosystem (Fig. 3F). For instance, in tumors, the T cell exclusion program of epithelial cells was closely associated with the Treg states, promoting immune escape and tumor progression[29,32]. The basal squamous state coincided with inflammatory states such as *CXCL9*[+] macrophage, *LAMP3*[+] DC, and mesenchymal-derived interferon states exclusively in tumor tissues, suggesting the intrinsic feature of basal squamous originated cancer types (HNSC and skin squamous cell carcinoma) that triggers the immune cell infiltration (Fig. 3F). In particular, multiple coincidences involving mesenchymal-derived interferon state with interferon states from other cell types and *LAMP3*[+] DC were identified (Fig. 3F). Its tumor-specific coincidence with the *LAMP3*[+] DC suggests that mesenchymal-derived interferon is necessary to initiate the tumor interferon signaling for the subsequent recruitment of immune cells and T cell priming.

## Characterization of *AKR1C1*[+] and *WNT5A*[+] inflammatory fibroblasts as distinct subtypes

Fibroblasts are a highly heterogeneous population with diverse functions such as collagen deposition, angiogenesis, and cytokine secretion that play a central role in forming the TME[33]. Fibroblasts promote inflammation and orchestrate the tissue microenvironment toward immunosuppression in the context of cancer[34], but the diversity of inflammatory fibroblasts has not been extensively explored in previous pan-cancer studies[30,31]. As we project mesenchymal cell collections upon the defined states, we identified multiple fibroblast subtypes that displayed immune-related gene expression (Fig. 4A). Among them, we noted the distinctions between *AKR1C1*[+] and *WNT5A*[+] expressing inflammatory fibroblasts, both of which coincided with PRR-induced activation state and shared similar cytokine gene expressions (*CXCL1/3/8*; Fig. 4A and Supplementary Fig. S12A). However, they significantly differed in terms of marker genes (*AKR1C1*, *FOSL1*, *LIF*, and *THAP2* vs. *WNT5A*, *GREM1*, *TNC*, and *MMP1*), tissue origins (normal vs. tumor), and organ preferences (Figs. 3E, 4A, B, and Supplementary Fig. S12B, C). To investigate whether these two states represent genuinely distinct subtypes of fibroblasts, we conducted a detailed examination of scRNA-seq datasets encompassing both tumor and normal tissues of the breast, colon, head and neck, and ovary. Tumor tissues expressed more chemokine genes (*CXCL1/3/8*) than normal tissues with the exception of the breast tissue. Simultaneously, the expression patterns of *AKR1C1* and *WNT5A* in these two types of inflammatory fibroblasts were distinct across different organs (Fig. 4C). BRCA and OV patients expressed both *AKR1C1* and *WNT5A*, while normal breast tissues expressed only *AKR1C1*. Intriguingly, colorectal cancer (CRC) and HNSC patients expressed *WNT5A* but not *AKR1C1* whereas the corresponding normal tissues showed opposite expression patterns (Fig. 4C).

To gain insight into how these inflammatory fibroblasts shape the TME, we performed ligand-receptor analysis to unravel distinct interactions of *AKR1C1*[+] and *WNT5A*[+] inflammatory fibroblasts with other cell types in BRCA, CRC, HNSC, and OV samples (Fig. 4D and Supplementary Fig. S12D, E). Discrete interaction patterns were identified between these two inflammatory fibroblasts. The *AKR1C1*[+] inflammatory fibroblast interacts with *CTSK*[+] macrophages (*CD44:SIGLEC15*), which are known to induce cancer progression and metastasis in multiple cancer types[35] (Fig. 4D and Supplementary Fig. S12D).

Moreover, the *AKR1C1*[+] inflammatory fibroblast strongly expresses *IL6* which interacts with *IL6R* expressed on DC1 and PRR-induced mo-DC, potentially contributing to tumor growth and therapeutic resistance[36,37]. We also identified an interrelation between *TNFRSF12A* (*AKR1C1*[+] inflammatory fibroblast) and *TNFSF12* (ILC3)[38]. In contrast, the *WNT5A*[+] inflammatory fibroblast interacts with cancer cells via *IL24:IL20RA* and *WNT5A:FZD5* pathways, which could promote cancer progression and chemoresistance[39,40] (Fig. 4D and Supplementary Fig. S12E). Furthermore, *WNT5A*[+] inflammatory fibroblast showed strong interactions with fibroblast populations including desmoplastic fibroblasts (*WNT5A:PTK7*, *WNT5A:ROR2*, and *GREM1:ACVR1*), with themselves in homotypic interactions (*WNT5A:PTK7* and *GREM1:ACVR1*), and *BMP4*[+] fibroblasts (*GREM1:ACVR1*) to mimic wound repair process to potentiate mesenchymal proliferation and migratory phenotype of cells through both autocrine and paracrine mechanism[41,42] (Fig. 4D and Supplementary Fig. S12E). Proangiogenic and proinflammatory engagement of *GREM1* from *WNT5A*[+] inflammatory fibroblast and *ACVRL1* from endothelial cells were also recognized[43]. Collectively, *WNT5A*[+] inflammatory fibroblasts secrete diverse ligands such as *WNT5A*, *GREM1*, and *IL24*, that potentiate proliferation, migration, and survival in numerous contexts, including tissue regeneration, inflammation, and cancer[41,44–46]. They interact with diverse cellular components such as cancer cells, fibroblasts, and endothelial cells to ultimately shape the pro-tumorigenic and core inflammatory TME. To locate and validate *WNT5A*[+] inflammatory fibroblast populations, we performed RNA single-molecule fluorescence in situ hybridization (smFISH) targeting *WNT5A*, *GREM1*, and *PDGFRA* in CRC and HNSC tissue samples. The presence of *WNT5A*[+] inflammatory fibroblasts was confirmed by the overlapping signals from the *WNT5A*, *GREM1*, and *PDGFRA* RNA probes in the desmoplastic stroma of cancer tissues (Fig. 4E and Supplementary Fig. S13, 14).

We hypothesized that distinct microenvironmental surroundings induce the divergent phenotypes of these two inflammatory fibroblasts. To examine their co-localization patterns, we analyzed spatial transcriptomics (BRCA, OV, and UCEC for *AKR1C1*[+] inflammatory fibroblast, and CRC and HNSC for *WNT5A*[+] inflammatory fibroblast). *AKR1C1*[+] inflammatory fibroblast significantly co-localized with cancer cells, neutrophil, *CTSK*[+] macrophage, DC1, and PRR-induced mo-DC (Fig. 4F, G and Supplementary Fig. S15A, B). In contrast, *WNT5A*[+] inflammatory fibroblast significantly co-localized with desmoplastic fibroblast, exhausted CD8[+] T cell, Treg, DC1, and PRR-induced mo-DC, highlighting its role in shaping the immunosuppressive TME (Fig. 4F, G and Supplementary Fig. S15A, B). In particular, *AKR1C1*[+] and *WNT5A*[+] inflammatory fibroblasts were mutually exclusive with desmoplastic fibroblast and neutrophil, respectively, further highlighting the distinctions between the two inflammatory fibroblasts (Fig. 4F, G and Supplementary Fig. S15A, B). The cellular interactions and spatial proximity of *WNT5A*[+] inflammatory fibroblasts with exhausted CD8[+] T cells and Tregs prompted us to investigate its association with immunotherapy treatment in pre/post-treated HNSC samples[47]. There was a trend toward an increase in *WNT5A*[+] inflammatory fibroblasts and desmoplastic fibroblasts after immunotherapy treatment, unlike other mesenchymal populations (Supplementary Fig. S15C). This indicates that *WNT5A*[+] inflammatory fibroblasts are potentially affected in the course of immunotherapy treatment. Collectively, although the two inflammatory fibroblasts express *CXCL1/3/8* in common, distinct organ/tissue preferences, cellular interaction patterns, and spatial proximities with other cell types suggest their differential role in forming an immune-evasive and pro-tumorigenic TME.

## Tumor-specific rewiring of interferon-enriched and pro-tumorigenic community

To depict multicellular ecosystems across tissue origins, we constructed an undirected network with co-occurrences of the cell states and identified diverse network communities within the tumor,

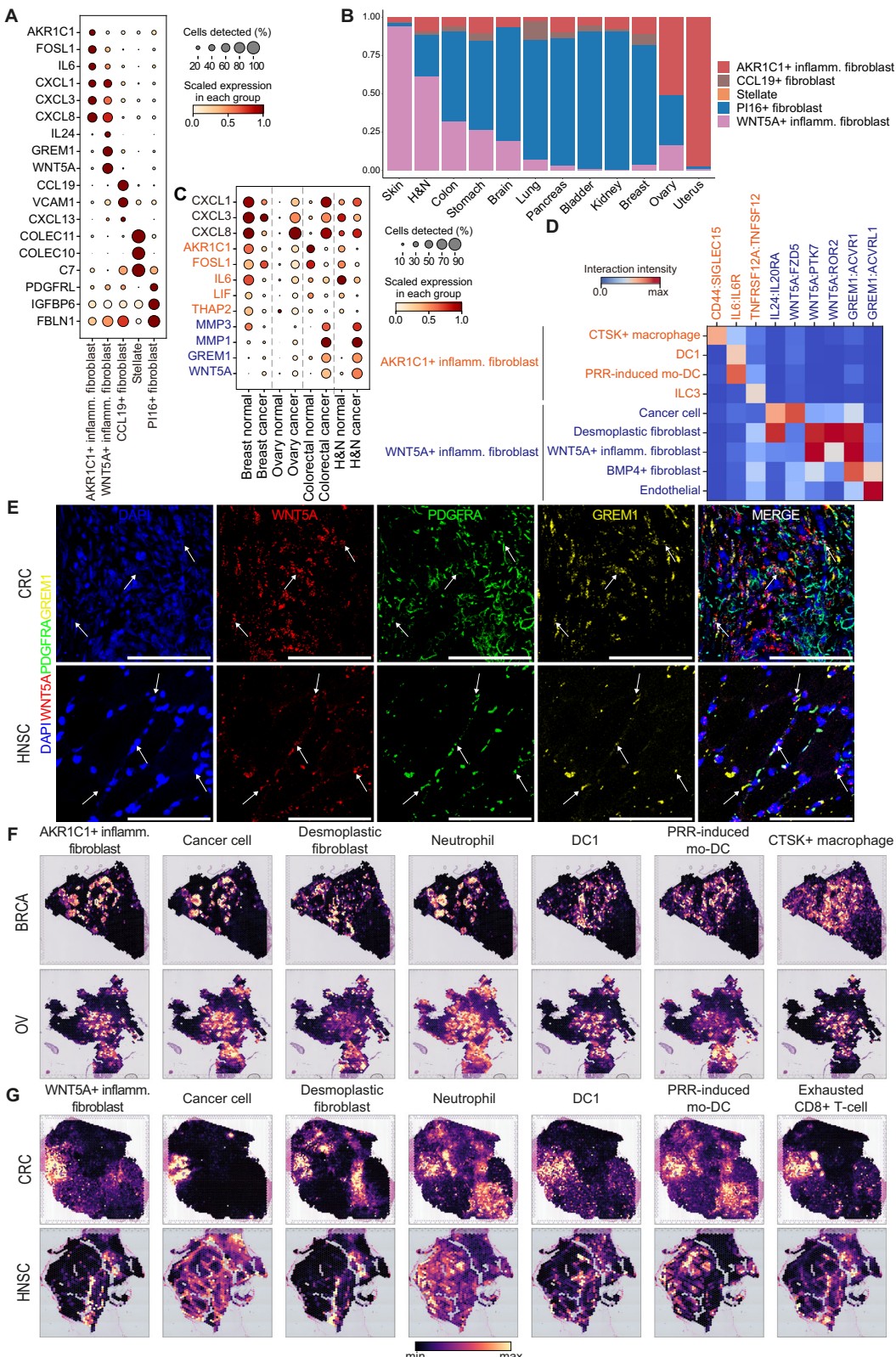

adjacent normal, and normal tissues (Fig. 5A–C). Remarkably, one tumor ecosystem was distinctly occupied by interferon states from various cell types within the tumor (interferon-enriched community; Fig. 5A). This community also contained well-known components of the TLS (*LAMP3*[+] DC, *CCL19*[+] fibroblast, and Tfh), particular DCs and macrophages (DC1, pDC, and *CXCL9*[+] macrophage), and antigen-presenting machinery states. In contrast, several immune cell states

included within the interferon-enriched community were scattered throughout the healthy normal network (Fig. 5B). Interestingly, the vicinity of *LAMP3*[+] DC with other immune cell states included in the interferon-enriched community (DC1, pDC, Langerhans cell, and Tfh) was identified in the adjacent normal network (Fig. 5C), implying that distinct configurations of cellular ecosystems even exist within the adjacent normal tissues compared to healthy normal tissues.

**Fig. 4 | Characterization of *AKR1C1*+ and *WNT5A*+ inflammatory fibroblasts.** **A** Dot plot of marker gene expressions of inflammatory fibroblast subtypes. Inflamm., inflammatory. **B** Distribution of inflammatory fibroblasts across organs where the y-axis represents the proportion of inflammatory fibroblasts in tumor tissues. H&N, head and neck; Inflamm., inflammatory. **C** Dot plot showing gene expression of *AKR1C1*+ and *WNT5A*+ inflammatory fibroblast marker genes in normal and tumor tissues of relevant organs. H&N, head and neck. **D** Ligand-receptor interactions of *AKR1C1*+ and *WNT5A*+ inflammatory fibroblasts with other cell types. The interaction intensity was calculated by multiplying the normalized expression values of ligands and receptors in each cell-cell pair. DC, dendritic cell; Inflamm.,

inflammatory; ILC3, type 3 innate lymphoid cells; mo-DC, monocyte-derived dendritic cell; PRR, pattern recognition receptor. **E** Representative (*n* = 3) images of in situ RNA smFISH detection of *WNT5A* (red), *PDGFRA* (green), and *GREM1* (yellow) in the desmoplastic stroma of CRC (top) and HNSC (bottom) tissues. scale bar: 100 μm. Magnification: 20X. Spatial co-localization patterns of (**F**) *AKR1C1*+ and (**G**) *WNT5A*+ inflammatory fibroblast with other cell types in relevant organs, with colors representing cell abundance. DC, dendritic cell; Inflamm., inflammatory; mo-DC, monocyte-derived dendritic cell; PRR, pattern recognition receptor. Source data are provided as a Source Data file.

---

We also identified another tumor-specific community occupied by pro-tumorigenic states, such as pEMT, epithelial T cell exclusion program, *SPP1*+ macrophage, and desmoplastic fibroblast (Fig. 5A). Tumor-enriched macrophages (*CTSK*+ and *C1QC*+ macrophage) and *WNT5A*+ inflammatory fibroblast states, which contribute to diverse aspects of tumorigenesis, were also found in the pro-tumorigenic community[40,48,49].

## Determination of immunotherapy-predictive cell states across cancer types

Having identified the diversity and dynamics of interferon-enriched and pro-tumorigenic communities among tumor, adjacent normal, and healthy normal tissues, we utilized those cell states to deconvolute the bulk transcriptomes of checkpoint inhibitor-treated samples in bladder cancer (BLCA), melanoma (MEL), RCC and our LC cohort (Fig. 5D). The clinical benefits of checkpoint blockade were highlighted by the exhausted CD8+ T cell, mesenchymal-derived interferon, *CXCL9*+ macrophage, *CD160*+ intraepithelial lymphocyte, Treg, DC1, *ISG15*+ macrophage, *XCL1*+/CD16+ NK cell, *IFIT1*+ interferon signaling, Tfh, GCB, *LAMP3*+ DC, pDC, CD16+ monocyte-derived macrophage, *CCL19*+ fibroblast, and plasma cell precursor states in the pan-cancer meta-analysis (Fig. 5E). Simultaneously, previously defined gene signatures such as the PD-L1 pathway, antigen-presenting machinery, and interferon signatures from various studies conferred favorable responses to immunotherapy. The cell states with significant predictive power were mostly components of the interferon-enriched community. Among the cell states favorable for immunotherapy, we confirmed the presence of interferon-expressing mesenchymal cells (*CXCL10* and *PDGFRA*) in the HNSC tissue sample using RNA smFISH (Supplementary Fig. S16).

In contrast, the desmoplastic fibroblast, osteoblast, mesothelium-derived fibroblast, and *CTSK*+ macrophage cell states were associated with poor responses to immunotherapy across the cohorts, many of which belonged to the pro-tumorigenic community (Fig. 5E). Considering that these pro-tumorigenic component states (i.e., desmoplastic fibroblast, osteoblast, and *CTSK*+ macrophage) negatively impact immunotherapy responses across different cancer types, alternative treatment strategies should be pursued for patients with pro-tumorigenic ecosystems. Notably, the majority of cell states predicting immunotherapy response did not significantly affect the prognosis across TCGA cohorts (Supplementary Figs. S17, 18).

Despite the heterogeneous effect of diverse cell states in immunotherapy, we questioned whether a common gene signature exists among responders. To identify genes that are differentially regulated regardless of organs between immunotherapy responders and non-responders for major cell types, we investigated immunotherapy cohorts of 4 cancer types at single-cell resolution (BRCA, LC, MEL, and RCC; Supplementary Data 5). In CD8+ T cells, *PDCD1*, *LAG3*, *CXCL13*, *CXCR6*, *VCAM1*, *CCDC141*, and *ZBED2* were commonly expressed in immunotherapy responders, while *ZNF80* was expressed in non-responders (Supplementary Fig. S19A and Supplementary Data 6). In macrophages, universal upregulation of *SCIN*, *OLFML3*, *PLD4*, *P2RY11*, and *SLAMF7* were identified in responders, while non-responders expressed *COLEC12*, *PROS1*, *CTSK*, *MOB3B*, *ADAMTSL4*, *TSPAN15*, and

*GPX3* (Supplementary Fig. S19A). Beneficial survival effects of universally expressed CD8+ T cell genes in immunotherapy responders were also validated in our LC cohort, indicating common gene expression programs in immunotherapy responders hold prognostic significance even in patients of different cancer types (Supplementary Fig. S19B).

## Systematic investigation of tumor ecosystems with spatial transcriptomics across multiple cancer types

To examine the spatial organization of the tumor ecosystem, we conducted a systematic analysis of spatial transcriptomics data across 11 different cancer types (*n* = 137, Fig. 6A). Given that TLS has previously been linked to favorable immunotherapy responses[12,13], we first aimed to derive a gene signature specific to TLS and also investigate spatial relationships between components associated with favorable responses to immunotherapy and pathologically defined TLS. To achieve this goal, we utilized the RCC spatial transcriptome data which contains pathologically defined TLS spots[50]. By performing differential expression analysis between TLS and non-TLS spots, we obtained a TLS signature that effectively distinguishes TLS from non-TLS spots (*p* = 1.5e−5; Fig. 6B and Supplementary Data 7). Our TLS signature also predicted favorable responses to immunotherapy in immunotherapy-treated cohorts (*p* = 1.4e−5; Fig. 6C and Supplementary Fig. S20A). Additionally, we utilized cell2location to deconvolute each spatial spot with single-cell transcriptome profiles and compared cell type abundances between TLS and non-TLS spots. We identified cell types that were significantly enriched in TLS, including Treg, plasma cell, CD16+/*XCL1*+ NK cells, Tfh, *LAMP3*+ DC, *CCL19*+ fibroblast, and ILC3 (Fig. 6D). It is noteworthy that some cell types, such as exhausted CD8+ T cell, *ISG15*+ macrophage, DC1, and pDC, lack spatial association with TLS, yet they exhibited favorable responses to immunotherapy similar to TLS-enriched cell types (Figs. 5E, 6D).

By evaluating the spatial co-localization patterns of TLS-enriched cell types in other cancer types, we confirmed the co-localization of TLS-enriched cell types including *LAMP3*+ DC, Treg, plasma cell, and *CCL19*+ fibroblast across multiple cancer types (Fig. 6E). Furthermore, we observed that our TLS signature effectively captures spots abundant with TLS-enriched cell types in various tumor samples of different cancer types (Fig. 6E). In contrast, pro-tumorigenic components exhibited distinct co-localization patterns (Supplementary Fig. S20B). Notably, desmoplastic fibroblasts and *SPP1*+ macrophages both co-localized with *WNT5A*+ inflammatory fibroblasts but were mutually exclusive with each other, indicating independent contributions to the pro-tumorigenic milieu. In summary, our study leveraged pan-cancer single-cell transcriptomes and spatial transcriptomes to characterize the intricate spatial organization of the TME.

## Discussion

In this work, we integrated a transcriptomic atlas comprising 4.9 million cells from 999 individuals across 30 cancer types, including both tumor and non-tumor tissues. Our analysis outlined hallmark gene signatures and systematically visualized the cell states that shape tumor-normal ecosystems. Furthermore, we elucidated distinctions between *AKR1C1*+ and *WNT5A*+ fibroblasts shaping the core

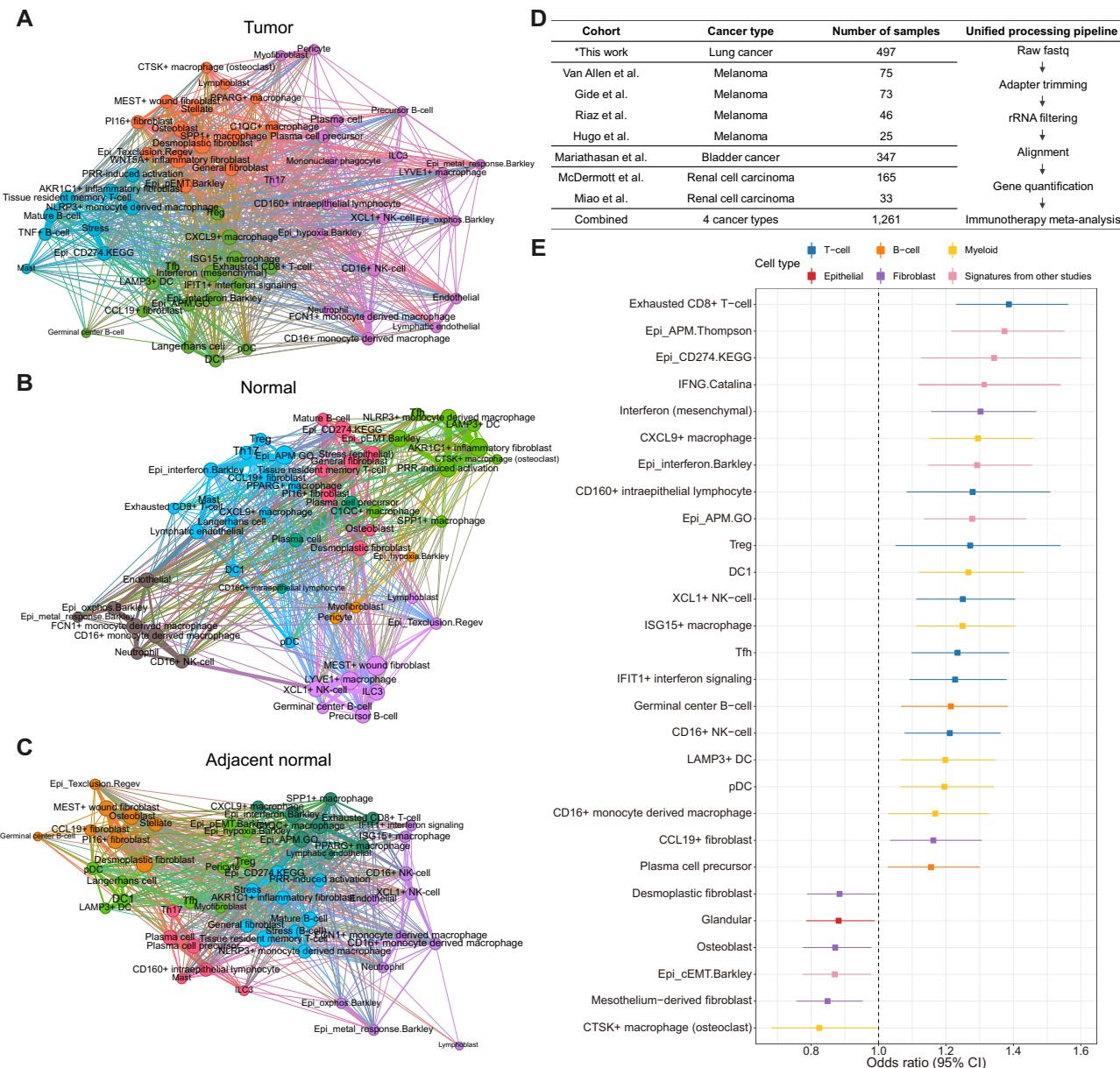

**Fig. 5 | Tumor-specific occurrence of interferon-enriched and pro-tumorigenic community and determination of immunotherapy-predictive cell states.** Co-occurrence network in the (**A**) tumor, (**B**) normal, and (**C**) adjacent normal tissues. The color of the nodes corresponds to the modularity community and the thickness of the edges corresponds to the magnitude of adjacency. DC, dendritic cell; EMT, epithelial-to-mesenchymal transition; ILC3, type 3 innate lymphoid cells; NK, natural killer cell; pDC, plasmacytoid dendritic cell; PRR, pattern recognition receptor; Texclusion, T cell exclusion program; Tfh, T follicular helper cells; Th17, T helper type 17; Treg, regulatory T cell. **D** Summary of the immunotherapy-treated bulk RNA-seq cohorts with response data used in this study. * indicates newly generated data. **E** Forest plot of immunotherapy-response predictive cell states and gene signatures from other studies through meta-analysis of

immunotherapy-treated bulk RNA-seq cohorts ($n = 1261$ patients). The x-axis represents the odds ratio, in which the dotted vertical line represents an odds ratio of 1, and the y-axis denotes cell states and previously defined gene signatures. For each cell state, rectangles and extended lines represent odds ratios and 95% confidence intervals, respectively, calculated through meta-analysis from logistic regression for clinical response across immunotherapy-treated cohorts. Only cell states that achieved statistical significance are depicted. Cell states with odds ratios greater than 1 are those associated with favorable responses to immunotherapy. The colors correspond to the cell type categories of each cell state. APM, antigen-presenting machinery; DC, dendritic cell; EMT, epithelial-to-mesenchymal transition; IFNG, interferon-gamma; NK, natural killer cell; pDC, plasmacytoid dendritic cell; Treg, regulatory T cell. Source data are provided as a Source Data file.

inflammatory TME. Diverse components of the interferon-enriched community including *LAMP3*⁺ DC, *CCL19*⁺ fibroblast, *CXCL9*⁺ macrophage, and mesenchymal-derived interferon states confer favorable responses to immunotherapy and form different ecosystems between tumor, adjacent normal, and healthy normal tissues. High-resolution annotation of comprehensive cell states from our pan-cancer single-cell atlas, in combination with spatial transcriptomics, enabled the categorization of cell types enriched in TLS and those that are not

within immunotherapy-favorable components, and the investigation of patterns that are highly reflective of complex tumor biology.

Although the gene expression divergence is mitigated during the malignant transformation of tissues in many cancer types[51], the evidence of the co-expressed gene modules in the TME at single-cell resolution is lacking. Herein, we aimed to identify hallmark gene expression programs in malignant tissues compared with normal tissues across various cell types, irrespective of the organs. Despite the

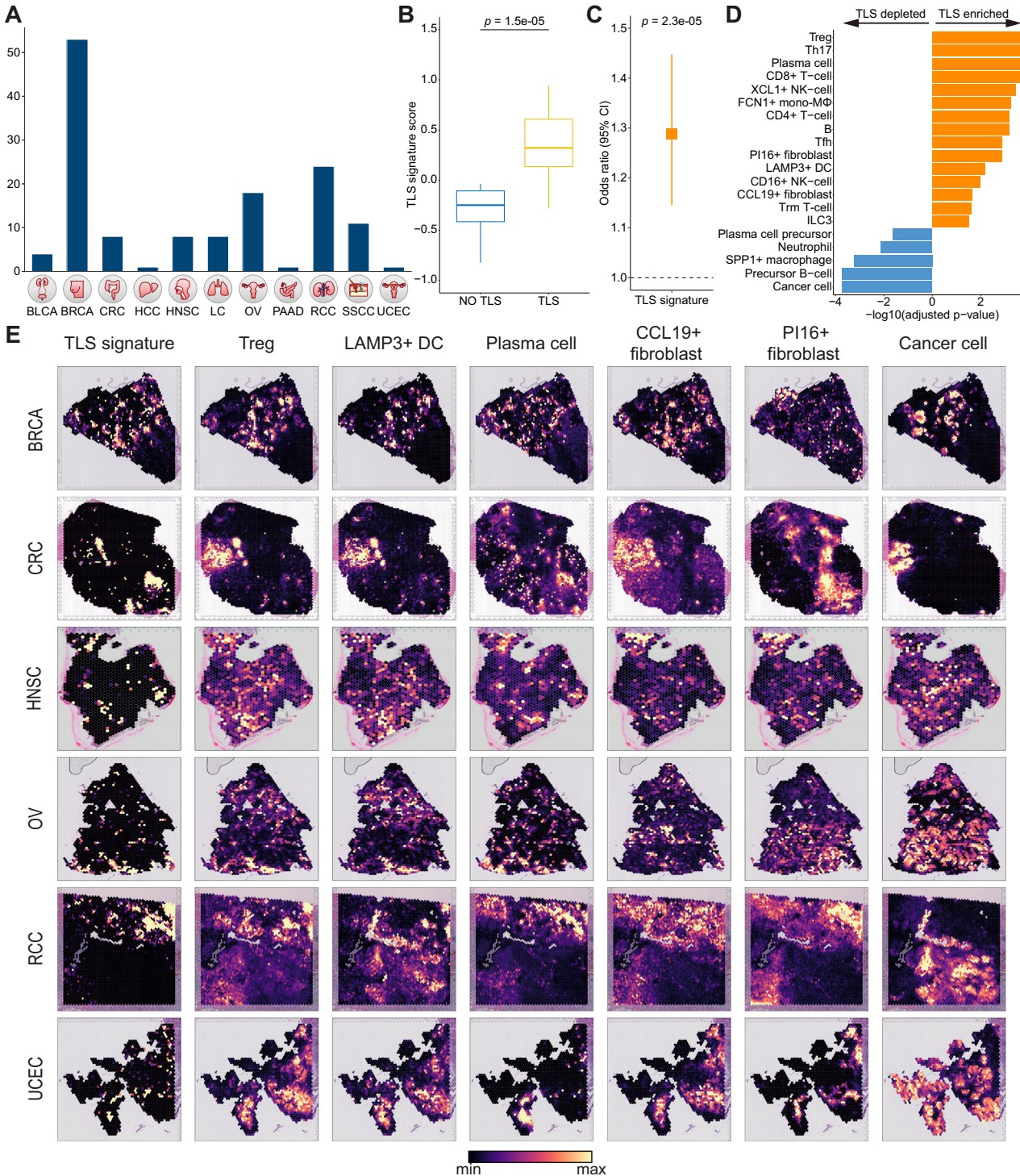

**Fig. 6 | Spatial transcriptome analysis of tumor ecosystems across multiple cancer types. A** Overview of pan-cancer spatial transcriptome analyzed in this study. **B** Boxplot comparing TLS signature scores between TLS and non-TLS spots. Statistical significance was calculated with the two-sided Wilcoxon rank sum test ($p = 1.5e^{-5}$, $n = 17$). In the box plot, the center line, upper box limit, lower box limit, and whiskers represent the median, first quartile, third quartile, and 1.5x inter-quartile range, respectively. **C** Predictive power of TLS signature for immunotherapy response in bulk RNA-seq cohorts treated with immunotherapy. The rectangle and extended lines represent the odds ratio and 95% confidence interval, respectively, calculated using TLS signature scores through meta-analysis from logistic regression for clinical response across immunotherapy-treated cohorts ($p = 1.4e^{-5}$,

$n = 1261$). Nominal two-sided $p$-values were obtained from the meta-analysis results of the logistic regression analysis. **D** Barplot of TLS-enriched cell types using TLS-labeled RCC spatial transcriptomes. The statistical significance of the enrichment or depletion was calculated using the two-sided Wilcoxon rank sum test and adjusted with the Benjamini-Hochberg method. Only the cell types reaching statistical significance are presented. DC, dendritic cell; ILC3, type 3 innate lymphoid cells; mono-MΦ, monocyte-derived macrophage; NK, natural killer cell; Tfh, T follicular helper cells; Th17, T helper type 17; Treg, regulatory T cell, Trm; Tissue-resident memory. **E** Spatial co-localizations of TLS signatures with diverse cell types across cancer types. DC, dendritic cell; Treg, regulatory T cell. Source data are provided as a Source Data file.

high heterogeneity embedded in the tissue ecosystems, our findings suggest that pan-organ gene expression programs exist across diverse cell types. These universal gene expression programs may account for the widespread application of immunotherapy (e.g., PD-1/PD-L1 inhibitors and CTLA4 inhibitors) in a variety of cancers, in contrast to the targeted therapy which has limited indications in specific cancer types. Comprehensive future studies are necessary to uncover the immuno-regulatory role of the TME hallmark genes, which may represent immune checkpoints or therapeutic targets in the TME of diverse cancer types.

Our study features a thorough investigation of the mesenchymal states in tumor and normal tissues. In particular, we identified *AKR1C1*[+] and *WNT5A*[+] inflammatory fibroblast states, which exhibited similar coincidence patterns with PRR-induced activation state and cytokine gene expression profiles. Although these inflammatory fibroblast populations have been previously recognized in several cancer types[40,52,53], our study provides a detailed exploration and uncovered the distinctions between these populations in terms of tissue and organ preferences, cellular interactions, and spatial co-localization patterns, thus underscoring the need for an integrative atlas. The *AKR1C1*[+] fibroblast state was characterized by genes such as *AKR1C1*, *CXCL8*, *CXCL2*, *CXCL3*, and *TNFAIP6* that are predictive of adverse responses to immunotherapy[54–58]. Therefore, targeting the *AKR1C1*[+] inflammatory fibroblast might offer new therapeutic insights, especially in OV and UCEC. Intriguingly, *WNT5A*[+] inflammatory fibroblast has been shown to interact with diverse TME components configuring inflammatory and pro-tumorigenic milieu via *WNT5A*, *GREM1*, and *IL24*[41,44–46]. Future studies that therapeutically target *WNT5A*[+] inflammatory fibroblasts should be pursued to mitigate deleterious consequences associated with these pro-tumorigenic fibroblasts. Moreover, myofibroblasts in our study exhibited relatively higher enrichment in normal tissues compared with the findings of the previous pan-cancer study[30]. It is worth noting that our study encompassed both tumor and normal samples from various organs, and the tissue enrichments of myofibroblasts represent a comprehensive outcome across all organs (Fig. 3E). Organ-wise analysis revealed myofibroblast enrichment in tumor tissues of the breast, pancreas, liver, and kidney (Supplementary Fig. S18C), suggesting a potential tumorigenic role of these fibroblasts in these organs[59–61]. However, tumor-normal enrichment of myofibroblasts in the colon and stomach was comparable, aligning with prior studies that have documented the presence of myofibroblasts in the normal intestine[62,63].

Various cell states arising from distinct cell types, many of which were components of the interferon-enriched community, were favorable for immunotherapy. While TLS has previously been identified as a predictor of immunotherapy response[12,13], we emphasize that both TLS-enriched and non-enriched cell types contribute positively to immunotherapy responses. Furthermore, our TLS signature, which effectively distinguishes TLS from non-TLS spots in RCC patients, predicts favorable responses to immunotherapy and also successfully co-localizes with TLS-enriched cell types in various cancer types. However, it remains to be confirmed whether TLS-enriched cell types are comparable in TLS-defined tissues of cancer types other than RCC. In addition, multiple states that confer adverse responses to the checkpoint blockade were also characterized, highlighting the need for alternative therapeutic options. To emphasize, although these immunotherapy-predictive cell states merely showed no prognostic significance on TCGA cohorts, immunotherapy showed differential treatment responses according to cell states for each cancer type, indicating that distinct mechanisms prevail in the course of immunotherapy treatment across cancer types.

Our study presents a distinctive perspective by constructing an extensive atlas that combines single-cell, spatial, and immunotherapy-treated bulk datasets, and by systematically comparing tumor and normal ecosystems across diverse organs. This allowed us to unravel hallmark gene signatures of the tumor-normal ecosystems, outline distinctions between *AKR1C1*[+] and *WNT5A*[+] inflammatory fibroblasts, and characterize TLS-enriched and non-enriched cell types among immunotherapy-favorable components. The pan-cancer tumor-normal single-cell meta-atlas presented in the study would provide essential insights into a deeper understanding of tumor-normal ecosystems and lay the groundwork for future studies in precision oncology. For the benefit of the research community, our atlas can be accessed through the Zenodo repository (DOI:10.5281/zenodo.10651059) and web portal (https://cellatlas.kaist.ac.kr/ecosystem/).

## Methods

### Ethics approval and consent to participate
Our immunotherapy cohort study was approved by the Institutional Review Board of Samsung Medical Center (SMC 2018-03-130). All patients enrolled in the study provided informed written consent.

### Our immunotherapy-treated lung cancer cohort
Histologically confirmed lung adenocarcinoma and lung squamous cell carcinoma patients, including previously reported cases[64], treated with either PD-1 or PD-L1 inhibitors were recruited. Clinical information of this cohort was collected from electronic medical records and tumor response was evaluated with Response Evaluation Criteria in Solid Tumors (v1.1). Progression-free survival (PFS) was defined as the time from the initiation of PD-1/PD-L1 inhibitors until the date of documented disease progression or death from any cause, whichever occurred first. Overall survival (OS) was defined as the time from the initiation of PD-1/PD-L1 inhibitors to death from any cause (Supplementary Data 8). The AllPrep DNA/RNA Mini Kit (Qiagen, USA) was used to purify RNA from formalin-fixed paraffin-embedded (FFPE) or fresh tumor samples. Subsequently, NanoDrop and Bioanalyzer (Agilent, USA) were used to measure the RNA concentration and purity. Library preparation was performed with either the TruSeq RNA Library Prep Kit v2 (Illumina, USA) or the TruSeq RNA Access Library Prep Kit (Illumina, USA), following the manufacturer's instructions. In total, we generated transcriptome data for 497 immunotherapy-treated LC patients.

### Single-cell data inclusion criteria and collection
For data selection, we first searched for relevant 10x scRNA-seq datasets to homogenize and reduce batch issues arising from different chemistries. Datasets were searched and downloaded from PubMed, Google Scholar, Gene Expression Omnibus, Single Cell Portal (https://singlecell.broadinstitute.org/single_cell), COVID-19 Cell Atlas (https://www.covid19cellatlas.org/), and Curated Cancer Cell Atlas (https://www.weizmann.ac.il/sites/3CA/).

The inclusion criteria were as follows: studies generated from 10x-genomics reagent kits and studies that include cancer, pre-cancerous, benign tumors, and normal samples. Among normal control samples, non-malignant tissues derived from cancer patients (annotated as adjacent normal) and tissues from healthy normal individuals (annotated as normal) were collected separately. Studies that only include sorted cells (e.g.,; CD45[+] sorting), fluid samples (e.g.; ascites, CSF, or PBMC), cell-line cultures, mouse studies, and studies generated from nuclei-seq were excluded. Altogether, a total of 104 scRNA-seq datasets comprising 1070 tumors, 493 normal (adjacent normal + healthy normal) samples, and approximately 4.9 million cells from 999 individuals, were collected and integrated (Supplementary Data 1).

### Single-cell RNA sequencing data analysis
The gene columns of each dataset were re-aligned according to the GRCh38 human reference genome (official Cell Ranger reference,

version 2020-A). For each dataset, cells with fewer than 1000 UMI counts and 500 detected genes were considered to be empty droplets and removed from the dataset. Cells with more than 7000 detected genes were considered potential doublets and removed from the dataset. The Scanpy (v.1.8.2) Python package was used to load the cell-gene count matrix and for analysis[65]. Clustering, annotation, and downstream analysis were performed using the tools in the Scanpy package complemented with some custom codes. Scrublet was used for doublet detection[66].

To reduce the computational burden and accelerate downstream analysis, a subset of cells was selected for each dataset with a Geometric sketch that mirrors the transcriptional diversity and preserves rare cell types[67] (Fig. 1B–D and Supplementary Fig. S1A).

## Cell type annotation and batch correction
We merged all tumor-normal scRNA-seq and divided the datasets to reduce the computational burden, visualize, and annotate cells. BBKNN was used as a batch effect correction algorithm to generate a connected graph structure[68], and after obtaining UMAP at a global scale, annotations were made based on cell type-specific marker genes (Supplementary Fig. S1B). Then, we examined and refined major cell type annotations for each dataset.

## Copy number variation inference for identification of malignant cells
Single-cell transcriptome-based large-scale copy number variation (CNV) of malignant cells was inferred using inferCNVpy (using the package available at https://github.com/icbi-lab/infercnvpy, v.0.4.2) with default window size and gencode v29 for genomic location reference. We employed *infercnvpy.tl.infercnv* to infer CNV and selected normal immune cells or fibroblasts as reference normal cells depending on each cancer type. Following dimensional reduction (*infercnvpy.tl.pca*) and clustering based on CNV profiles (*infercnvpy.tl.leiden*), cells were visualized on a CNV UMAP (*infercnvpy.tl.umap*) and CNV scores were calculated using *infercnvpy.tl.cnv_score*. Putative malignant cells were defined based on two criteria: (i) Formation of separate clusters, a known property of malignant cells[69], and (ii) higher CNV scores compared with known normal cell types (normal epithelial, fibroblasts, or immune cells based on each cancer type).

## AND-gating algorithm of differential expression of genes for identification of hallmark gene signatures
We applied an AND-gating algorithm to extract tumor-enriched or immunotherapy-favorable gene signatures for each cell type across different cancers[70] (Supplementary Fig. S2A). Cells from a specific organ were subsetted and differential expression analysis was performed to identify genes that were (i) upregulated in the cell type of interest compared with other cell types (log$_2$ fold-change >0) and (ii) highly expressed in the cell type of interest in tumor tissues (or immunotherapy responders) compared with normal tissues (or immunotherapy non-responders; log$_2$ fold-change >0.5 and adjusted $p < 0.05$). Genes that satisfied both conditions were retained to create tumor-specific/immunotherapy-favorable gene signatures. The $p$ values were calculated using the two-sided $t$ test on log-normalized gene matrices and adjusted with the Benjamini-Hochberg method (Python packages scipy.stats v.1.10.0 and statsmodels.stats v.0.13.5). After obtaining the gene signatures derived from the AND-gating for each cell type and organ, we integrated the gene signatures from multiple organs to identify hallmark gene signatures for each cell type.

## Biological annotation of the hallmark gene signatures of tumor-normal ecosystems
We utilized Enrichr[71] to annotate the biological functions of the tumor-specific hallmark gene signatures across diverse cell types. We employed GO terms from MsigDB Hallmark 2020, GO Biological Process 2023, and GO Molecular Function 2023, and only the terms with an adjusted $p < 0.05$ were considered as significant (Fig. 2C).

## NMF pre-processing and visualization
After cell type annotation, we performed NMF for each individual tissue separately, accounting for each cell type category and tissue origin, to generate cell states that contribute to the heterogeneity within each individual. Starting from the log-normalized centered expression matrix of all genes, the negative values were set to zero. The *sklearn.decomposition.NMF* method was applied with default parameters as implemented in the scikit-learn python package v1.0.2. Considering that NMF requires a K parameter that influences the results, we ran NMF using different values (K = 5,6,7,8,9), thereby generating 35 modules for each individual.

Next, we applied a graphical approach to cluster and visualize the NMF modules. First, we max-normalized and merged all modules derived from each cell type and converted it to an anndata object. After highly variable gene selection, we removed low-quality modules (modules with an NMF weight less than 10−20 or greater than 150−170, depending on each cell type). Dimensional reduction using principal components and UMAP visualization was performed, and small module clusters with less than 150 modules were removed. Leiden clustering was then performed and a list of the top 50 genes was derived for each cluster. We then removed clusters enriched with either ribosomal protein genes or mitochondrial-encoded genes, composed of NMF modules from a single study, and suspected to reflect doublet cells or the soup effect based on high similarity to the expression profile from another cell type (e.g.; T cells genes in mesenchymal NMF modules).

## Automation of doublet or soup effect cluster removal
To identify and remove NMF module clusters (cell states) indicating doublet cells or the soup effect, we built an algorithm for automating doublet or soup effect cluster detection. In detail, we identified the 2 organs (only 1 organ if there were more than 2-fold differences of NMF module counts between the two predominant organs) that compose the majority of the NMF cluster of interest. We then subsetted the corresponding organs in our geometric sketched anndata of our tumor-normal meta-atlas. With the top 50 genes derived from the cluster, we scored each cell type category in the subsetted anndata using the *sc.tl.score* function in the Scanpy package. If the scores were greater than 0.2 in a different cell type (e.g., T cells) compared with the cell type of interest (e.g., mesenchymal), the cluster was defined as a doublet or soup effect cluster and was subsequently removed. Clusters with higher mesenchymal scores were not discarded from epithelial cell states to prevent the EMT state from being removed.

## Defining and annotating cell states
After visualizing NMF modules and clustering the states, the top 50 genes with weighted averages were identified for each state. For overlapping genes between cell states, we orthogonally assigned the genes to the state with a higher NMF weighted average (Supplementary Data 2). Azimuth (https://azimuth.hubmapconsortium.org/), The Human Protein Atlas (https://www.proteinatlas.org/), and Enrichr[71] were used as main references to annotate cell states. Also, to validate our cell states, we compared our states to the gene signatures derived from other studies with the Pearson correlation (Supplementary Fig. S4).

## Construction of reference components with the cell states
To assess the correspondence between cell states and cell subtypes, we projected cells utilizing cell-type specific cell state profiles as reference components[15]. For cell states from each cell type category, orthogonal genes with weighted averages were identified for each

state (see Defining and annotating cell states). Then, cell cycling and cell states derived from ambient RNAs or doublets were removed, and the genes constituting the remnant states were selected as variable genes in the log-normalized scRNA-seq dataset. Reference components were constructed by performing matrix multiplication between the scRNA-seq anndata and the cell state weighted averages (RCA=anndata.X.dot(cell state weighted average)). These reference components were used to replace the principal components, and subsequently, BBKNN was performed, using the dataset as a batch key[68]. Then, final cell type annotations were made based on cell type-specific marker genes (Fig. 3B, D and Supplementary Figs. S6, and 7).

## Measuring cell state score distributions and coincidence analysis

To identify which cell states are enriched in the cells of each individual, we scored each individual with the orthogonal genes of the cell states using the *sc.tl.score_genes* function to measure cell state scores. For 8 cancer types that constitute the majority of the pan-cancer atlas (BRCA, CRC, HCC, HNSC, LC, OV, PAAD, and RCC), we calculated the mean scores for each individual, and Pearson correlation was performed between cell states to measure the coincidence. For each coincidence between cell states, the adjacency was calculated with the WGCNA package (v.1.71)[72], and the Circos plot was depicted with the circlize package[73] (v.0.4.15). The thickness of lines in the Circos plot corresponds to the adjacency between cell states.

## The ratio of observed to expected of cell states

To quantify the tissue or organ preference of the cell states, we calculated the ratio of observed to expected (Ro/e). To quantify the tissue Ro/e, we created a $3 \times 2$ contingency table by counting the occurrence of tissue origins (i.e.; normal, adjacent normal, and tumor) of NMF modules from the cell state of interest and others. To simultaneously consider the tissue and organ origins when calculating the Ro/e, we first extracted NMF modules of a cell state, determined proportions of organ origins within these modules, and filtered out those derived from organs constituting less than 3%. Then, for each tissue origin, a contingency table was created by counting the occurrence of the organ origins of NMF modules from the cell state of interest versus other cell states. An expected number was derived using chi-square analysis and Ro/e was calculated as $\log_2 \frac{observed}{expected}$. We considered a cell state enriched or depleted in a specific tissue/organ if Ro/e > 0 or Ro/e < 0, respectively.

## Ligand-receptor interaction analysis

To understand the functional characteristics of *AKR1C1*+ and *WNT5A*+ inflammatory fibroblasts, we investigated the potential cellular interactions between *AKR1C1*+ and *WNT5A*+ inflammatory fibroblasts with other cell types using cell-cell interaction inference tools such as CellPhoneDB (v.3.1.0)[74], focusing on gene expression programs specific to these two inflammatory fibroblasts. The interaction intensity was calculated by multiplying the normalized expression values of ligands and receptors in each cell-cell pair.

## RNA smFISH

RNA smFISH was performed on FFPE tissue sections obtained from desmoplastic stromas in patients with CRC and HNSC. We used RNAscope® probes targeting human *WNT5A* (AD604921, 1:1500, Opal 570), *GREM1* (AD312831-C2, 1:1500, Opal 690), and *PDGFRA* (AD604481-C3, 1:3000, Opal 520) to identify *WNT5A*+ inflammatory fibroblasts, and *CXCL10* (AD311851, 1:1500, Opal 570) and *PDGFRA* (AD604481-C2, 1:3000, Opal 520) to identify interferon-expressing fibroblasts. Five-μm thick FFPE tissue sections were deparaffinized with xylene and subsequently processed with RNAscope® Multiplex Fluorescent Reagent Kit Assay, following the manufacturer's instructions. Fluorescence images were acquired with a ZEISS LSM 980 confocal

microscope using FITC (Opal 520), TRITC (Opal 570), Cy5.5 (Opal 690), and DAPI channels.

## Survival analysis using bulk transcriptome

To evaluate the prognostic value of cell states and hallmark signatures in each cancer type, we performed survival analysis with TCGA RNA-seq data. Upper-quartile normalized FPKM data was collected from UCSC Xena across 28 cancer types[75]. TCGA clinical data (OS) was obtained from data provided by the TCGA Pan-Cancer clinical data resource[76]. Enrichments of cell states and hallmark signatures were calculated for each TCGA primary cancer sample using the single-sample gene set enrichment analysis (ssGSEA) function implemented in the Corto package (v.1.1.10)[77]. Patients were grouped into depleted and enriched groups based on the average cell state score of the analyzed samples. The Kaplan−Meier curves were plotted with a *ggsurvplot* function and the log-rank test was performed to quantify statistical significance. The Benjamini-Hochberg method was applied to correct multiple testing. To evaluate the prognostic significance of cell states in relevant organs, we determined the Ro/e filtering threshold for each cell type using the following formula:

$$\text{Ro/e filtering threshold} = \text{mean(Ro/e)} - 1\text{s.d.(Ro/e)} \qquad (1)$$

We identified cell states as rare within an organ if the tumor-derived Ro/e values for those cell states within that organ did not exceed the filtering threshold. This prevented the deconvolution of rare cell states that are not relevant to the survival analyses of specific organs.

## Network construction with cell states

We constructed an undirected network with cell states to visualize their co-occurrence patterns. When constructing each tissue network, we utilized cell states that are identified in the corresponding tissue origin (e.g., *WNT5A*+ inflammatory fibroblast in tumor network only). After calculating the co-occurrence of cell states, the adjacency matrix was calculated with the WGCNA package (v.1.71)[72]. The adjacency values of the cell state pair with a *p* value greater than 0.05 were set to 0 to minimize spurious effects. Then, the adjacency matrix was imported into gephi (v.0.10.1) to construct a connected network[78]. We conducted community detection with default parameters, colored the nodes with modularity class, and enlarged the nodes with the average weighted degree. We selected ForceAtlas2 for graph embedding.

## Collection and processing of immunotherapy cohorts' transcriptome data

We collected bulk transcriptome of immunotherapy-treated cohorts (4 cancer types, 8 cohorts). Transcriptomic data of the LC cohort was generated by Samsung Medical Center (our LC cohort, $n = 497$; See Our immunotherapy-treated lung cancer cohort). For MEL cohorts, Van Allen et al. ($n = 75$)[79], Gide et al. ($n = 73$)[80], Riaz et al., ($n = 46$)[81], and Hugo et al. ($n = 25$)[82] cohorts were gathered. Along with the BLCA cohort by Mariathasan et al. ($n = 347$)[83], RCC cohorts including McDermott et al. ($n = 165$)[84] and Miao et al. ($n = 33$) were also collected[85]. For all cohorts, the raw FASTQ files were obtained and processed in a unified pipeline (Fig. 5D). First, the adapter sequences in the FASTQ file were trimmed out by Trimmomatic (v.0.39)[86]. To filter out rRNA, SortMeRNA (v.2.1b) was used[87]. The filtered reads were aligned to the hg38 reference genome by the STAR aligner (v.2.7.6a) in two-pass basic mode with gencode annotation (v.35)[88,89]. The aligned reads were then sorted by samtools (v.1.7) and the read counts were calculated by HTSeq (v.0.12.4)[90,91]. To quantify gene expression, the read counts were normalized in the TPM value.

## Analysis of immunotherapy cohorts

ssGSEA with gseapy (v.0.10.8) was performed to score cell state per sample and the normalized enrichment score was used in the analysis[92]. For our pan-cancer analysis, only samples with both response and survival data were included; patients who had durable clinical benefits (complete response, partial response, stable disease with PFS > 6 months or OS > 1 year) were classified as responders, and others as non-responders. We then conducted a meta-analysis to identify the associations between the clinical response to immunotherapy and cell states in multiple cohorts. First, the scaled signature score was fitted to the clinical response with logistic regression for each cohort. The calculated estimates and standard errors were then pooled across cohorts using the *metagen* function in the meta package[93]. Finally, considering the heterogeneity between studies, the random effect model was established to estimate the effect of the cell states. From the random effect model, the overall estimate, standard error, and p-values were obtained.

## Pan-cancer spatial transcriptome analysis

Spatial transcriptome analysis of 137 cancer datasets across 11 cancer types was performed with cell2location[94] using default parameters to quantify the spatial distribution of cell types (Fig. 6A). For each spatial transcriptome cancer type, we employed the respective cancer scRNA-seq datasets from our pan-cancer single-cell atlas as a reference. Then, we employed spot-wise Pearson correlation with an estimated abundance of cell types to quantify spatial colocalization patterns, analogous to previous studies[95,96]. A high positive Pearson correlation indicated that two cell types exhibited similar spatial distributions, while a negative Pearson correlation suggested distinct spatial distributions between the two cell types.

## Construction of TLS gene signature and identification of TLS-enriched cell types

To create a TLS signature and pinpoint cell types enriched in TLS using the RCC spatial transcriptome data with information of pathologically defined TLS spots[50]. A pseudo-bulk matrix was compiled for both TLS and non-TLS spots in each sample, and subsequent differential expression analysis was performed with PyDESeq2 (v.0.4.3)[97]. From genes exhibiting an adjusted $p < 0.01$, we selected the top 50 genes with the highest $\log_2$ fold-change values (Supplementary Data 7). Enrichment of TLS gene signature was quantified with the *sc.tl.score* function of the Scanpy package for each spot, enabling the identification of spots enriched with the TLS signature in spatial transcriptome tissues[65]. To identify TLS-enriched cell types, we compared cell abundances between TLS and non-TLS spots with the Wilcoxon signed-rank test (Python packages scipy.stats v.1.10.0) and characterized cell types that are enriched in TLS (Fig. 6B).

## Reporting summary

Further information on research design is available in the Nature Portfolio Reporting Summary linked to this article.

## Data availability

The scRNA-seq and spatial transcriptome datasets analyzed in this study are provided in Supplementary Data 1, 3, and 5, along with their accession codes and links. The processed scRNA-seq and spatial transcriptome data are available at Zenodo repository (DOI:10.5281/zenodo.10651059)[98]. Processed immunotherapy-treated lung cancer cohort data generated in this study have been deposited in the Gene Expression Omnibus repository under accession code GSE218989. Raw sequencing data of the immunotherapy-treated lung cancer cohort have been deposited in the European Genome-phenome Archive (EGA) under controlled access with accession number EGAD50000000469. Data requests for academic or intellectual purposes will be reviewed by the data access committee, and are expected to be responded within 4 weeks. Our dataset can be interactively visualized at https://cellatlas.kaist.ac.kr/ecosystem/. Source data are provided with this paper.

## Code availability

The codes used for data analysis are available from the Zenodo repository (https://doi.org/10.5281/zenodo.10651059)[98].

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

## Acknowledgements
This work was supported by the National Research Foundation (NRF) funded by the Korean government (MSIT) (NRF-2022R1A4A5028131, NRF-2021M3A9I4024447, NRF-2021R1C1C1010094, and NRF-2020R1A2C3006535), Future Medicine 2030 Project of the Samsung Medical Center (SMX1240011), and HR21C019803. This research was supported by a grant from the MD-PhD/Medical Scientist Training Program (Junho Kang) and Korea Health Technology R&D Project (HR21C0198) through the Korea Health Industry Development Institute (KHIDI), funded by the Ministry of Health & Welfare, Republic of Korea. Jong-Eun Park was supported by the POSCO Science Fellowship. The authors acknowledge the Korea Research Environment Open Network (KREONET) service and the usage of the Global Science Experimental Data Hub Center (GSDC) provided by Korea Institute of Science and Technology Information (KISTI). This publication is part of the Human Cell Atlas – www.humancellatlas.org/publications/.

## Author contributions
J.K.C and J.E.P conceived and supervised the study. J.H.Kang and D.H.A collected and analyzed the datasets, J.H.Kang and J.H.L performed computational analysis, contributed to the interpretation of data, and prepared the first draft of the manuscript. S.H.L and H.C generated the immunotherapy-treated LC data, and A.Y.K and H.J.A collected patient tissue samples. J.H.Kang, J.H.L, S.L, and S.K performed experiments, J.H.Kang, J.H.L, J.H.Kwon, J.H.A, and S.Y.K made contributions to figure generation, and M.Y.B and J.H.Kang created the web portal. All authors revised the draft critically for important intellectual content and gave final approval for the final manuscript version to be published.

## Competing interests
The authors declare no competing interests.
