## [Peer Review File · Nature Communications]

REVIEWER COMMENTS

Reviewer #1 (Remarks to the Author): Expert in computational cancer genomics, tumour microenvironment, and scRNA-seq analysis

The paper presents a pan-cancer single-cell transcriptomic atlas across 30 cancer types aiming at characterizing the cellular diversity and complexity of various tumor microenvironments (TME). The authors used multiple analyses to examine the molecular and cellular features of millions of single cells, comparing tumors and normal tissues, as well as evaluating their spatial distribution and interactions. The manuscript describes different cell states that comprise the TME and evaluates their impact on tumor progression, organ specificity, and response to immunotherapy. The paper provides a good resource for a comprehensive and high-resolution view of tumor ecosystems.

This work addresses an important topic. It has a clear broad goal to improve our understanding of inter- and intra-tumoral heterogeneity and explore new ways for better cancer detection and therapy. The amount of work done is also recognizable. However, the paper lacks a clear narrative. The amount of information the authors provide in this manuscript makes it hard to follow. Readers will struggle to distil what discoveries are more important, which are less, what findings are novel and which were published before.

Here are a few additional specific concerns:

- 1) In the abstract, the authors wrote, "Notably, we discovered AKR1C1+ and WNT5A+ inflammatory fibroblasts". However, other publications have described these populations (e.g., PMID: 26573806, 34175688, 36459995).
- 2) The main and supplemental figure legends are terribly shy in their information, making it very hard to understand in detail what each panel describes. In some figure legends, necessary information is completely missing (e.g., the axes on the circos plot, Fig. 3C).
- 3) On the other hand, some figures include too many details and do not communicate results effectively (Fig. S5E-F)
- 4) There are multiple occasions when the authors describe one result yet refer the reader to multiple data sources. For example, lines 208-210 refer to a main figure, a supplemental figure and a table. This is cumbersome to read and follow.
- 5) Many details concerning crucial analyses are missing in the method section (e.g., ligand-receptor analysis). Readers will find it difficult to reproduce results.
- 6) The author should separate malignant from non-malignant epithelial cells in their different analyses (e.g., by CNV inference)
- 7) It is unclear if the number of cells in each sample affects the results. That is, signals are missing due to the absence of specific cells. Including the cell numbers will make it easier to evaluate the importance of

different cell states. For example, I will find it difficult to trust results of survival analyses from bulk RNA-seq after the deconvolution of cell states representing rare populations.

8) On a similar note, in the survival analyses, using the 25th and 75th percentiles rather than a median split, for example, means that 50% of the patients are being omitted. This inflates the real significance of the different cell states to affect prognosis.

9) I found at least one instance where genes mentioned in the text cannot be found in the reference. For example, in lines 160-161, the authors note the genes CXCL14, COL4A5, GRK5, and LRRC45 in the pancreas, ovary, kidney, and lung, respectively. However, I could not find entries for these genes in Table S4.

10) It is unclear why some results refer to only a few of the many interrogated cancer types. For example, Fig. 2 includes only six cancer types. Breast, which is their most abundant, is missing.

11) Lines 195-202 refer to epithelial cell states, yet the supplementary figures include cell types that are not epithelial (e.g., neuron, oligodendrocyte, Schwann cell, astrocyte)

12) Fig. 4 and Fig. 6 express high frequencies of specific cell populations in the spatial transcriptomics data. This seems to contrast their much lower frequencies in the scRNAseq atlas (Fig. 3). Can the authors address this?

13) There are no details about how spatial proximity was calculated and what accounts as significant. The method section starting at line 897, is very vague. It needs to be clarified how these correlations were obtained.

14) In Fig. S16E, it is not clear how "Brain vascular endothelial" is significantly associated with survival in a completely unrelated organ system (i.e., kidney cancer). Without an explanation, this may make other results questionable too.

15) It needs to be explained how the composite gene signature was derived (In Line 383), and there are no details of what genes compose the signature.

16) It is not clear to me why in Fig S14, the results are consistently more significant in the "in-house" lung cancer data compared with the other cancer types. The authors should address this.

Reviewer #2 (Remarks to the Author): Expert in cancer bioinformatics, tumour immune microenvironment, single-cell and spatial transcriptomics

The manuscript by Kang and Lee et al. presents a comprehensive study on the tumor-immune-stroma ecosystem of TME, encompassing single-cell datasets, spatial omics, and immunotherapy cohorts. It presents both originally generated data by the authors and publicly available datasets with data integration, re-annotation, and curation at a global level. The authors found that specific cell states, such as CXCL10 fibroblasts, and TLS-associated CXCL9 Mo, are associated with response to immunotherapy. They also reported hallmark programs across cancer types in the context of therapy response and/or organ-specific biology.

This study is absolutely stunning considering the scale of datasets used as single-cell, spatial, therapy-related, and cancer types. The amount of work the author have conducted to ensure data and methods are described and shared is also phenomenal. With the recognition of several pan-cancer single-cell studies already published in the past two years, this study presents its unique angle of analyzing both tumor and normal samples, as well as in silico validation from single-cell to spatial datasets. However, one major challenge of such a large-scale study is the technical heterogeneity that could impact the cell annotation, data distribution, etc., and hence may affect the conclusion of specific cell states in the context of organ, cancer type, or therapy response.

1. how were the different technologies handled, e.g., 5' or 3' 10X, among others? How about 10x and smartseq data? Due to the intrinsic design of those technologies, some transcripts are detected in one technology but not the other. Therefore, the results based on gene expression similarity may be confounded by differences in technologies, in addition to biology differences e.g., cancer types or organ/tissue type.

2. various batch correction and integration techniques have been applied to combine data sets for analysis, how to prove data were not overcorrected?

3. malignant cells intrinsically form clusters by patient (as shown in many other scRNAseq cancer studies), due to patient specific TME factors (e.g., somatic mutations). How are those patient-specific differences taken into account when looking for common patterns between cancer types?

4. NMF method was extensively discussed in the literature, with pros and cons. When it comes to using NMF to find shared patterns between groups of cells, the field has been controversial in whether “the negative values were set to 0” is a proper preprocessing step before applying NMF – it can almost be perceived as removing data to fit the assumption of NMF, which can be dangerous. Recognizing there are several big single-cell papers published in high-impact journals utilizing NMF for this purpose, it might not be reasonable to ask for re-analyzing data using an alternative method. However, authors may consider at least demonstrating that they can reproduce the same results using methods other than NMF – such as scCoGAPS (PMID: 31121116), which does not require removing data to fit the assumptions. scCoGAPS is only mentioned here as an example, other tools may serve this purpose, I will leave it to the authors' preference to address this question. Authors may consider demonstrating this in major cancer types if not all.

5. TLS has been previously reported to be associated with immunotherapy response. However, its prevalence is cancer-specific, for example, they are more prevalent in melanoma and lung, but less in bladder, or renal cell carcinoma, especially mature TLS. In addition to all datasets combined, have

authors investigated in which cancer type(s) the immunotherapy response predictive power of TLS is detected (was it MEL, LC?)? How is this TLS signature different from known immunotherapy biomarkers, such as IFNG signature, or are they confounded? Is TLS a surrogate of an IFNG-associated TME (e.g., highly correlated), or do these two signatures each act as independent biomarkers of immunotherapy response?

6. lastly, it will be important for the paper to provide raw data objects or files after authors' QC, but before applying integration, normalization, or batch correction methods. That way, users have the liberty to apply their own methods, which will broaden the impact of the data from this paper.

Reviewer #3 (Remarks to the Author): Expert in cancer genomics, scRNA-seq, tumour microenvironment, and immunotherapy

please see my attached file

By integrating 30 cancer types with over 4.9 million unsorted single-cell transcriptomic data, JE. Kang *et al* aimed to profile the tumor microenvironment at a higher resolution with large samples size. They further explored the signature molecular and cellular states, which may contribute to immunotherapy response outcome. The study has an impressive data integration and are organized carefully. However, the following points should be considered or explained.

1. The overall UMAP pattern is slightly different between the figure provided in manuscript and the website (<https://cellatlas.kaist.ac.kr/ecosystem/>). Some clusters seem to be manually deleted in the manuscript (see below). For instance, on the website, several epithelia clusters are located far from the others, especially one cluster is located on the top (see below), please explain why?

2. Multiple subclusters were divided including a lot of novel annotated. Indeed, some of these “novel” clusters have been annotated with different names in previous reports, particularly functional annotated. For instance, PI16+ fibroblast is more likely stem cell niche. Therefore, the “novel” clusters, which have been reported, should be carefully re-annotated.
3. The definition of inflammatory CAF in Figure 4 is confusion. For instance, PI16+ fibroblast is more likely stem cell niche, which is highly enriched in normal/adjacent samples. Therefore, this cluster are more likely normal fibroblast rather than CAF, how can it be annotated as inflammatory CAF? To exclude such confusion, it is important to illustrate the proportion of normal/adjacent and tumor for each cluster.
4. TLS community is an interesting topic in this study. However, the information is very limited and confusing. Some issues should be considered or explained: 1. Most of the cell types included in Figure 5A-5C are not specific to TLS, how to distinguish the TLS located cells and whether some cell types are TLS-specific? 2. TLS is barely observed in normal tissue, but some cell types highlighted in this study (e.g., CXCL9+ macrophage) are still presented in normal, thus can't be claimed as TLS-specific. 3. It

is better to construct a TLS signature and labeled in spatial transcriptome.

5. according to Fig. 6C, ISG15+ macrophage ranks the highest correlation with composite signature, but is lacking in the spatial demonstration in Fig. 6B.
6. this study claimed several novel cell subclusters, some of which are small in population according to the proportion analysis. IHC or multiplexed immunofluorescence (mIF) is needed to establish its real existence. In addition, some cell subtypes (e.g., CXCL9+ macrophage) were considered to be located in TLS. Since TLS is much easier to be revealed through IHC/mIF rather than spatial transcriptome, it is better to illustrate the localization of these cells in TLS structure through experimental approach with several examples.
7. the repository dataset in Zenodo is restricted. Although “Reviewers are allowed to access our repository” but when I request the access, name and email address should be provided. In addition, GSE218989 can't be found in GEO database, and the follow-up information of the lung cancer cohort is missing, which may be provided as supplementary table.
8. Line 842, the algorithm is missing.

REVIEWER COMMENTS

Reviewer #1 (Remarks to the Author): Expert in computational cancer genomics, tumour microenvironment, and scRNA-seq analysis

The paper presents a pan-cancer single-cell transcriptomic atlas across 30 cancer types aiming at characterizing the cellular diversity and complexity of various tumor microenvironments (TME). The authors used multiple analyses to examine the molecular and cellular features of millions of single cells, comparing tumors and normal tissues, as well as evaluating their spatial distribution and interactions. The manuscript describes different cell states that comprise the TME and evaluates their impact on tumor progression, organ specificity, and response to immunotherapy. The paper provides a good resource for a comprehensive and high-resolution view of tumor ecosystems.

This work addresses an important topic. It has a clear broad goal to improve our understanding of inter- and intra-tumoral heterogeneity and explore new ways for better cancer detection and therapy. The amount of work done is also recognizable. However, the paper lacks a clear narrative. The amount of information the authors provide in this manuscript makes it hard to follow. Readers will struggle to distil what discoveries are more important, which are less, what findings are novel and which were published before.

→ We thank the reviewer for the encouraging comments and for acknowledging the importance of our study. We conducted a comprehensive review of our manuscript and relevant literature to emphasize novel findings that set them apart from previously reported results and included an extra paragraph in the discussion section. Specific details can be found in the point-by-point responses and are marked in red throughout the manuscript. We would also like to kindly inform that our study has been included as part of the Human Cell Atlas projects (HCA-96), and we have acknowledged this information in the Acknowledgement section.

Here are a few additional specific concerns:

1) In the abstract, the authors wrote, "Notably, we discovered AKR1C1+ and WNT5A+ inflammatory fibroblasts". However, other publications have described these populations (e.g., PMID: 26573806, 34175688, 36459995).

→ We really appreciate this valuable comment. For the diversity of cancer fibroblasts, it is well known that there are inflammatory types of cancer-associated fibroblasts, defined by the expression of inflammatory cytokines such as *CXCL1/3/8* (PMID:33622705). However, the heterogeneity within the inflammatory fibroblasts across various cancer types has not been thoroughly investigated.

In our analysis, we classified two distinct inflammatory fibroblast subpopulations that share common expressions of *CXCL1/3/8*, as illustrated below. However, these two populations significantly differed in terms of marker genes (*AKR1C1*, *FOSL1*, *LIF*, and *THAP2* vs. *WNT5A*, *GREM1*, *TNC*, and *MMP1*), tissue origins (normal vs. tumor), organ preferences (breast, ovary, and uterus vs. head and neck, and colon), cellular interactions,

and spatial co-localization patterns. Consequently, we recognized the need for separate annotations for these two inflammatory fibroblast populations, which we have designated as *AKR1C1*⁺ and *WNT5A*⁺ inflammatory fibroblasts, as *AKR1C1* and *WNT5A* account for the distinctions between them.

[Dotplot of marker gene expression patterns between *AKR1C*⁺ and *WNT5A*⁺ inflammatory fibroblasts (Figure 4A)]

[Marker gene expressions of *AKR1C1*⁺ and *WNT5A*⁺ inflammatory fibroblasts in UMAP (Figure S12C)]

At the same time, several prior studies highlighted the utility of *AKR1C1* and *WNT5A* as a marker for cancer fibroblast populations as the reviewer suggested. Hence, we have revised the relevant part of the abstract to highlight the analysis conducted in this study as follows:

“Our atlas characterized distinctions between inflammatory fibroblasts marked by *AKR1C1* or *WNT5A* in terms of cellular interactions and spatial co-localization patterns.”

While the presence of *AKR1C1* and *WNT5A* expressing fibroblast populations have been reported previously, to the best of our knowledge, our study is the first to delineate the differences between these two inflammatory fibroblast populations that share *CXCL1/3/8* in common. We have also added a paragraph in the discussion section as follows:

“Our study features a thorough investigation of the mesenchymal states in tumor and normal tissues. In particular, we identified *AKR1C1*⁺ and *WNT5A*⁺ inflammatory fibroblast states, which exhibited similar coincidence patterns with PRR-induced activation state and cytokine gene expression profiles. Although these inflammatory fibroblast populations have been previously recognized in several cancer types (reference PMID: 26573806, 34175688, 36459995), our study provides a detailed exploration and uncovered the distinctions between these populations in terms of tissue and organ preferences, cellular interactions, and spatial co-localization patterns, thus underscoring the need for an integrative atlas.”

2) The main and supplemental figure legends are terribly shy in their information, making it very hard to understand in detail what each panel describes. In some figure legends, necessary information is completely missing (e.g., the axes on the circos plot, Fig. 3C).

→ We apologize for the lack of detailed information. We now have updated figures (e.g.: circos plot of Fig 3) and provided comprehensive information in both main and supplementary figure legends, as depicted below. We have made additional improvements and have highlighted them in red within both the main and supplementary figure legends to convey crucial information.

[Circos plot of co-occurrence between different cell states (Figure 3F)]

Figure 3F: Circos plot depicting the co-occurrences between the cell states in normal (green) and tumor (yellow) tissues. The length of the arcs represents the sum of co-occurrences with other TME components in adjacency. A longer arc means more frequent co-occurrence with other cell states. Basal sq, basal squamous state. DC, dendritic cell; Tex, exhausted CD8⁺ T-cell; T-exclusion, T-cell exclusion program; Treg, regulatory T cell.

[Pan-cancer immunotherapy forest plot (Figure 5E)]

Figure 5E: Forest plot of immunotherapy-responsive cell states and gene signatures from other studies through pan-cancer meta-analysis. The dotted vertical line represents an odds

ratio of 1. Cell states with odds ratios greater than 1 are those associated with favorable responses to immunotherapy. The colors correspond to the cell type categories of each cell state. Only cell states that achieved statistical significance are depicted. APM, antigen-presenting machinery; DC, dendritic cell; EMT, epithelial-to-mesenchymal transition; IFNG, interferon-gamma; NK, natural killer cell; pDC, plasmacytoid dendritic cell; Treg, regulatory T cell.

[Identification of TLS-enriched cell types with RCC spatial transcriptomes (Figure 6B)]

Figure 6B: Evaluation of TLS-enriched cell types using TLS-labeled RCC spatial transcriptomes. The statistical significance of the enrichment or depletion was calculated using the Wilcoxon rank sum test and adjusted with the Benjamini-Hochberg method. Only the cell types reaching statistical significance are presented. DC, dendritic cell; ILC3, type 3 innate lymphoid cells; mono-derived MΦ, monocyte-derived macrophage; NK, natural killer cell; Tfh, T follicular helper cells; Th17, T helper type 17; Treg, regulatory T cell.

3) On the other hand, some figures include too many details and do not communicate results effectively (Fig. S5E-F)

→ We apologize for the unnecessarily detailed information. As the reviewer pointed out, previous supplementary figures S5E–S5F do not communicate with the results effectively. We now have modified the figures for more efficient visualization of our data as exemplified in the figure below and utilized updated supplementary figures of Ro/e according to the organ of origin (now available in Figures S16 and S17) in survival analysis to deconvolute cell states that do not represent rare populations (please refer to comment 7 and 14).

[Organ of origin Ro/e analysis of B cell states (Figure S16E)]

4) There are multiple occasions when the authors describe one result yet refer the reader to multiple data sources. For example, lines 208-210 refer to a main figure, a supplemental figure and a table. This is cumbersome to read and follow.

→ We acknowledge the reviewer's valid point regarding the referring of one result to multiple data sources repeatedly. For efficient readership, we have made efforts and updated our manuscript as demonstrated below. We have made additional enhancements and marked them in red throughout the manuscript, which will assist readers in following our figures and tables.

Previous lines 208–210:

“Notably, we classified and annotated diverse cell states of mesenchymal origin, including the *CCL19*⁺ fibroblast (*CCL19* and *CXCL13*), *P116*⁺ fibroblast (*MFAP5* and *IGFBP6*), myofibroblast (*ACTA2* and *MYH11*), desmoplastic fibroblast (*LRRC15* and *MMP11*), and fibroblast states exclusive to specific organs that potentially reflect organ-specific functions (Figures 3C–3D and S11).”

5) Many details concerning crucial analyses are missing in the method section (e.g., ligand-receptor analysis). Readers will find it difficult to reproduce results.

→ We apologize for the lack of detailed information. We have updated analyses in detail in the methods section as below and also provided our analysis codes in the Figshare repository (DOI:10.6084/m9.figshare.24312331). Please kindly access through a private link for reviewers (<https://figshare.com/s/a51b42a7ab60b31aba2f>).

Receptor-ligand interaction analysis

To understand the functional characteristics of *AKR1C1*⁺ and *WNT5A*⁺ inflammatory fibroblasts, we investigated the potential cellular interactions between *AKR1C1*⁺ and *WNT5A*⁺ inflammatory fibroblasts with other cell types using cell-cell interaction inference tools such as CellPhoneDB (PMID: 32103204), focusing on gene expression programs specific to these two inflammatory fibroblasts.

Pan-cancer spatial transcriptome analysis

Spatial transcriptome analysis of 137 cancer datasets across 11 cancer types was performed with cell2location (PMID: 35027729) using default parameters to quantify the spatial distribution of cell types. For each spatial transcriptome cancer type, we employed the respective cancer scRNA-seq datasets from our pan-cancer single-cell atlas as a reference (Figure 6A). Then, we employed spot-wise Pearson correlation with an estimated abundance of cell types to quantify spatial colocalization patterns, analogous to previous studies (PMID: 36922516, <https://doi.org/10.1101/2022.12.22.521551>). A high positive Pearson correlation indicated that two cell types exhibited similar spatial distributions, while a negative Pearson correlation suggested distinct spatial distributions between the two cell types.

Survival analysis using bulk transcriptome

To evaluate the prognostic value of cell states and hallmark signatures in each cancer type, we performed survival analysis with TCGA RNA-seq data. Upper-quartile normalized FPKM data was collected from UCSC Xena across 28 cancer types (PMID: 32444850). TCGA clinical data (OS) was obtained from data provided by the TCGA Pan-Cancer clinical data resource (PMID: 29625055). Enrichments of cell states and hallmark signatures were calculated for each TCGA primary cancer sample using the ssGSEA function implemented in the Corto package (v.1.1.10) (PMID: 32232425). Patients were grouped into depleted and enriched groups based on the average cell state score of the analyzed samples. The Kaplan-Meier curves were plotted with a ggsurvplot function and the log-rank test was performed to quantify statistical significance. The Benjamini-Hochberg method was applied to correct multiple testing. To evaluate the prognostic significance of cell states in relevant organs, we determined the Ro/e filtering threshold for each cell type using the following formula:

$$\text{Ro/e filtering threshold} = \text{mean}(\text{Ro/e}) - 1 \text{ s.d}(\text{Ro/e})$$

We identified cell states as rare within an organ if the tumor-derived Ro/e values for those cell states within that organ did not exceed the filtering threshold. This prevented the

deconvolution of rare cell states that are not relevant to the survival analyses of specific organs.

Biological annotation of the gene signatures

We utilized Enrichr (PMID: 27141961) to annotate the biological functions of the TME hallmark gene signatures across diverse cell types. We employed GO terms from MsigDB Hallmark 2020, GO Biological Process 2023, and GO Molecular Function 2023, and only the terms with an adjusted p-value < 0.05 were considered as significant (Figure 2C).

6) The author should separate malignant from non-malignant epithelial cells in their different analyses (e.g., by CNV inference)

→ As pointed out by the reviewer, we utilized infercnvpy (reference: <https://github.com/icbi-lab/infercnvpy>) to infer malignant cells as shown below:

[Example: CNV inference using infercnvpy in ovarian cancer datasets]

With these results, we characterized recurring upregulated hallmark genes in cancer cells, analyzed receptor-ligand interaction patterns between malignant cells and *WNT5A*⁺ inflammatory fibroblasts, and examined their spatial co-localization patterns. We have incorporated these updates into our figures and manuscript, as illustrated in the figure below.

[Gene ontology analysis of TME hallmark genes of diverse cell types using Enrichr (Figure 2C)]

[Interaction of *AKR1C1*⁺ and *WNT5A*⁺ inflammatory fibroblast with other cell types (Figure 4D)]

[Spatial co-localization patterns of *AKR1C1*⁺ and *WNT5A*⁺ inflammatory fibroblast with other cell types (Figure 4F–4G)]

Additionally, we have made the CNV inference results available on the Figshare repository so that it can be utilized as a resource for readers (DOI:10.6084/m9.figshare.24312331). Please kindly access through a private link for reviewers (<https://figshare.com/s/a51b42a7ab60b31aba2f>).

7) It is unclear if the number of cells in each sample affects the results. That is, signals are missing due to the absence of specific cells. Including the cell numbers will make it easier to evaluate the importance of different cell states. For example, I will find it difficult to trust results of survival analyses from bulk RNA-seq after the deconvolution of cell states representing rare populations.

→ To address this concern, we have employed our Ro/e figures, categorized by the organ of origin (previous supplementary figures S5E–S5F), to avoid the deconvolution of cell states representing rare populations. To exclude infrequent cell states across various cancer types, we determined the Ro/e filtering threshold for each cell type using the following formula:

$$\text{Ro/e filtering threshold} = \text{mean}(\text{Ro/e}) - 1 \text{ s.d.}(\text{Ro/e})$$

We identified cell states as rare within an organ if the tumor-derived Ro/e values for those cell states within that organ did not exceed the filtering threshold, as exemplified below.

[Ro/e analysis showing tissue/organ preferences of mesenchymal cell states (Figure S17C)]

This approach aimed to prevent the deconvolution of rare cell states that are not relevant to the survival analyses of specific organs (e.g., alveolar fibroblasts in colon cancer & glioblastoma). Consequently, we have updated the survival analysis results accordingly as illustrated below.

Overall survival

[Prognostic impact of mesenchymal states in relevant TCGA cancer types (Figure S17D)]

8) On a similar note, in the survival analyses, using the 25th and 75th percentiles rather than a median split, for example, means that 50% of the patients are being omitted. This inflates the real significance of the different cell states to affect prognosis.

→ We agree with the reviewer's opinion that employing the 25th and 75th percentiles excludes the middle half of the patients and may exaggerate the true significance of the cell states on prognosis. Consequently, we have re-evaluated the survival analyses using a median split and subsequently updated the manuscript, methods, and figures accordingly, as exemplified below.

Overall survival

[Prognostic impact of T/NK states in relevant TCGA cancer types (Figure S16B)]

Survival analysis using bulk transcriptome

To evaluate the prognostic value of cell states and hallmark signatures in each cancer type, we performed survival analysis with TCGA RNA-seq data. Upper-quartile normalized FPKM data was collected from UCSC Xena across 28 cancer types (PMID: 32444850). TCGA clinical data (OS) was obtained from data provided by the TCGA Pan-Cancer clinical data resource (PMID: 29625055). Enrichments of cell states and hallmark signatures were calculated for each TCGA primary cancer sample using the single-sample gene set enrichment analysis (ssGSEA) function implemented in the Corto package (v.1.1.10) (PMID: 32232425). Patients were grouped into depleted and enriched groups based on the average cell state score of the analyzed samples. The Kaplan-Meier curves were plotted with a ggsurvplot function and the log-rank test was performed to quantify statistical significance. The Benjamini-Hochberg method was applied to correct multiple testing.

9) I found at least one instance where genes mentioned in the text cannot be found in the reference. For example, in lines 160-161, the authors note the genes CXCL14, COL4A5, GRK5, and LRRC45 in the pancreas, ovary, kidney, and lung, respectively. However, I could not find entries for these genes in Table S4.

→ We apologize for the omitted data in Table S4. In response to the reviewer's comment, we intended to include this data during the revision. However, when analyzing the survival analyses with a median split (refer to comment #8), we discovered that organ-specific gene signatures did not exhibit prognostic significance. Consequently, we made the decision to remove the corresponding text.

10) It is unclear why some results refer to only a few of the many interrogated cancer types. For example, Fig. 2 includes only six cancer types. Breast, which is their most abundant, is missing.

→ We thank the referee for highlighting this crucial point. Given that the differential expression gene (DEG) analysis was conducted on patients with both tumor and paired adjacent normal tissue samples, it did not encompass breast tissue samples in the analyses. In response to the reviewer's suggestion to incorporate the most prevalent cancer type, we have made adjustments and performed DEG analysis between tumor and normal tissues (rather than paired adjacent normal tissues). Consequently, we have updated our analysis results within the manuscript, figures, and supplementary tables as shown below.

[Hallmark gene signatures of diverse cell types (Figures 2A–2B)]

11) Lines 195-202 refer to epithelial cell states, yet the supplementary figures include cell types that are not epithelial (e.g., neuron, oligodendrocyte, Schwann cell, astrocyte)

→ We thank the reviewer for raising this valid concern. We acknowledge that referring to those cell types as 'epithelial' is inaccurate. Instead, it would be more precise to label them as 'epithelial and neural cells.' Therefore, we have opted to describe these cell populations as 'epithelial and neural cells' and have accordingly revised the manuscript and figure legends.

12) Fig. 4 and Fig. 6 express high frequencies of specific cell populations in the spatial transcriptomics data. This seems to contrast their much lower frequencies in the scRNAseq atlas (Fig. 3). Can the authors address this?

→ To address this important point, we examined the cell type proportion of our single-cell atlas and cell type abundances in spatial datasets in cancer types with more than 10 samples as shown below.

Cell Proportion in single-cell RNA-seq

Abundance proportion in spatial transcriptome

BRCA

OV

SSCC

RCC

[Cell type proportion in single-cell atlas and cell abundance in spatial transcriptomes]

Considering the significant “inter-tumoral” complexity variation among patients, the specific cell type abundance depicted in each spatial transcriptomic dataset reflects cell type abundance within a single patient. In contrast, the cell type proportions in the single-cell atlas represent the average cell type proportion at a population level. For instance, in the ovarian cancer patient shown in Fig. 4F (shown below), neutrophils appear to be highly abundant. However, the overall cell type abundance of neutrophils in spatial transcriptomic datasets from ovarian cancer patients is low, consistent with the cell type proportions observed in the single-cell atlas. Therefore, we conclude that the high frequency of specific cell populations in certain patients can be attributed to the inter-tumor heterogeneity observed among cancer patients and that an individual case does not necessarily represent the entire population. It is important to note that such inter-tumor heterogeneity may contribute to variations in the proportion of cell populations between the single-cell atlas and spatial transcriptome.

[Spatial co-localization patterns of *AKR1C1*⁺ inflammatory fibroblast with other cell types in breast cancer and ovarian cancer (Figure 4F)]

13) There are no details about how spatial proximity was calculated and what accounts as significant. The method section starting at line 897, is very vague. It needs to be clarified how these correlations were obtained.

→ We appreciate this valuable feedback. As we described in response to comment #5, we have added our spatial co-occurrence calculation methodology with more details in the manuscript. Additionally, we have incorporated citations to studies that have employed similar spatial co-occurrence analyses as follows:

Pan-cancer spatial transcriptome analysis

Spatial transcriptome analysis of 137 cancer datasets across 11 cancer types was performed with cell2location (PMID: 35027729) using default parameters to quantify the spatial distribution of cell types. For each spatial transcriptome cancer type, we employed the respective cancer scRNA-seq datasets from our pan-cancer single-cell atlas as a reference (Figure 6A). Then, we employed spot-wise Pearson correlation with an estimated abundance of cell types to quantify spatial colocalization patterns, analogous to previous studies (PMID: 36922516, <https://doi.org/10.1101/2022.12.22.521551>). A high positive Pearson correlation indicated that two cell types exhibited similar spatial distributions, while a negative Pearson correlation suggested distinct spatial distributions between the two cell types.

Also, please note that we have deposited the spatial analysis source code and associated datasets in the Figshare repository (DOI:10.6084/m9.figshare.24312331). Please kindly access through a private link for reviewers (<https://figshare.com/s/a51b42a7ab60b31aba2f>).

14) In Fig. S16E, it is not clear how "Brain vascular endothelial" is significantly associated with survival in a completely unrelated organ system (i.e., kidney cancer). Without an explanation, this may make other results questionable too.

→ Similar to comment #7, we have used our organ of origin Ro/e figures (updated Figures S16A, S16C, S16E, S17A, and S17C) to filter out rare or irrelevant cell state populations. We have then updated the survival analysis accordingly, as exemplified below.

[Ro/e analysis showing tissue/organ preferences of mesenchymal cell states (Figure S17C)]

Overall survival

[Prognostic impact of mesenchymal states in relevant TCGA cancer types (Figure S17D)]

15) It needs to be explained how the composite gene signature was derived (In Line 383), and there are no details of what genes compose the signature.

→ We apologize for the insufficient information provided. Initially, the composite gene signature was formulated by appending genes associated with immunotherapy-favorable cell states to identify immunotherapy hotspots within spatial datasets. In response to another reviewer's suggestion that it would be more advantageous to develop a Tertiary Lymphoid Structure (TLS) signature and apply it to the spatial transcriptome (refer to comment #4 of Reviewer 3), we have now constructed a TLS signature by incorporating cell state genes from the respective cell types significantly enriched within the TLS (Supplementary Table S7). Accordingly, we have updated the manuscript and figures, and provided a detailed description in the methods section as follows:

Construction of TLS gene signature

To create a TLS signature, we first aimed to pinpoint cell types enriched to TLS using the RCC spatial transcriptome data and the information from pathologically defined TLS spots (PMID: 35231421). By comparing cell abundance between TLS and non-TLS spots with the Wilcoxon signed-rank test (Python packages `scipy.stats` v.1.10.0), we characterized the cell types that are enriched in TLS (Figure 6B). Subsequently, by compiling the cell state genes from these TLS-enriched cell types, we established a TLS signature (Table S7). Enrichment of TLS gene signature was quantified with the 'sc.tl.score' function of Scanpy for each spot to identify TLS-signature enriched spots in spatial transcriptome tissues (PMID: 29409532).

16) It is not clear to me why in Fig S14, the results are consistently more significant in the "in-house" lung cancer data compared with the other cancer types. The authors should address this.

→ In our effort to address this critical comment, we scrutinized all individual cohorts in terms of patient characteristics. As a result, it was brought to our attention that the number of responders in the individual cohorts examined in Fig S14, except for our lung cancer cohorts, was limited (see the table below).

Cohort	Cancer type	Number of patients	Number of responders
David	Renal cell carcinoma	165	84
Gide	Melanoma	73	50
Hugo	Melanoma	26	14
Maria	Bladder cancer	347	104
Miao	Renal cell carcinoma	33	13
Riaz	Melanoma	46	21
SMC	Lung cancer	497	225
Van Allen	Melanoma	75	30

[Patient characteristics of the immunotherapy cohorts included in this work]

We suspected that this might have caused limited statistical power for the individual cancer types, that is, melanoma, renal cell carcinoma, and bladder cancer, in our analysis of Fig S14. When an adequate number of responders is achieved by combining all cohorts, excluding our lung cancer cohort, we observe a pattern similar to Figure 5E and Fig S14. Consequently, we obtained an equivalent number of cell states that significantly predict the response to immunotherapy, as depicted in the forest plot below.

[Predictive power of cell states in immunotherapy-treated cohorts]

Due to the limited number of responders in the bladder, melanoma, and renal cell carcinoma cohorts, which may have introduced confounding results in Fig S14, we have decided to remove Fig S14 and its corresponding content from the manuscript.

Reviewer #2 (Remarks to the Author): Expert in cancer bioinformatics, tumour immune microenvironment, single-cell and spatial transcriptomics

The manuscript by Kang and Lee et al. presents a comprehensive study on the tumor-immune-stroma ecosystem of TME, encompassing single-cell datasets, spatial omics, and immunotherapy cohorts. It presents both originally generated data by the authors and publicly available datasets with data integration, re-annotation, and curation at a global level. The authors found that specific cell states, such as CXCL10 fibroblasts, and TLS-associated CXCL9 Mo, are associated with response to immunotherapy. They also reported hallmark programs across cancer types in the context of therapy response and/or organ-specific biology.

This study is absolutely stunning considering the scale of datasets used as single-cell, spatial, therapy-related, and cancer types. The amount of work the author have conducted to ensure data and methods are described and shared is also phenomenal. With the recognition of several pan-cancer single-cell studies already published in the past two years,

this study presents its unique angle of analyzing both tumor and normal samples, as well as in silico validation from single-cell to spatial datasets. However, one major challenge of such a large-scale study is the technical heterogeneity that could impact the cell annotation, data distribution, etc., and hence may affect the conclusion of specific cell states in the context of organ, cancer type, or therapy response.

→ We thank the reviewer for acknowledging the importance and the implications of this work. We have made efforts to address the points raised by the reviewer, and now feel that our paper has been improved considerably thanks to the constructive comments. We would also like to kindly inform that our study has been included as part of the Human Cell Atlas projects (HCA-96), and we have acknowledged this information in the Acknowledgement section.

1. how were the different technologies handled, e.g., 5' or 3' 10X, among others? How about 10x and smartseq data? Due to the intrinsic design of those technologies, some transcripts are detected in one technology but not the other. Therefore, the results based on gene expression similarity may be confounded by differences in technologies, in addition to biology differences e.g., cancer types or organ/tissue type.

→ We appreciate the reviewer's insightful observation. To construct a comprehensive pan-cancer tumor-normal atlas, we systematically curated 10X-based datasets, encompassing both 5' and 3' chemistries, while excluding datasets generated by smart-seq or nuclei-seq methods. Among the collected datasets, we carefully filtered out those that did not integrate well with others during the initial cell type annotation by checking individual datasets. Acknowledging the potential variability in transcript detection between different chemistries, we sought to find the batch-free latent space by performing a patient-specific non-negative matrix factorization (NMF) approach for each cell type and searched for robust factors to utilize them for data projection. This facilitated the identification of recurring gene expression programs, which we referred to as "cell states". As we have run the analyses per sample, our factors were free of batch effects. This approach resulted in the identification of a total of 100 distinct cell states that collectively constitute the tumor-normal ecosystems. Importantly, none of these cell states exhibited exclusive detection or bias toward a particular chemistry (3' or 5') as illustrated below. Furthermore, we harnessed these cell type-specific cell state profiles as a reference for embedding, enabling the identification of cell types that faithfully correspond to their respective cell states.

[Distribution of cell states according to different chemistries]

2. various batch correction and integration techniques have been applied to combine data sets for analysis, how to prove data were not overcorrected?

→ We thank the referee for raising this point which is also linked with the reviewer's first comment described above. To combine and analyze a vast amount of datasets, we initially conducted NMF to identify recurring gene expression programs (cell states) that constitute the tumor and normal ecosystems (references: PMID 37258682, 35931863). Subsequently, using these cell type-specific cell states as references, we employed Reference Component Analysis (RCA), a method known for accurate clustering of single cells (reference: PMID 28319088). Finally, to integrate the extensive single-cell RNA-seq data, we implemented an integration pipeline that includes BBKNN (reference: PMID 31400197), a tool demonstrated to efficiently remove technical batch variations in an atlas-level benchmarking study (reference: PMID 34949812). With these identified cell types that represent each cell state, we inferred cellular interaction patterns and deconvoluted spatial transcriptomes to examine spatial co-localizations among the cell types of interest.

To assess the issue of over-correcting the data, we performed the following analysis. Firstly, as a simple sanity check, we confirmed the enrichment of cell type-specific cell state scores in each annotated cell type, which show very good correspondence.

[Difference in cell state score in the cell type of interest versus other cell types - Mesenchymal & B, **** p<0.0001]

[Difference in cell state score in the cell type of interest versus other cell types - T/NK & Myeloids, **** p<0.0001]

Additionally, when comparing our RCA with other batch correction methods (as illustrated below), the silhouette score of RCA exceeded that of both Harmony and SCVI for every cell type. This suggests that RCA successfully clustered cell types at an atlas level.

Furthermore, we annotated immune cells in our atlas using Celltypist (PMID: 35549406) and conducted a comparison of our RCA with other batch correction methods (as depicted below). We noticed that RCA achieved a higher silhouette score in myeloid and B cells, although it was slightly lower in T/NK cells. In summary, these findings suggest that our atlas has not undergone excessive correction.

3. malignant cells intrinsically form clusters by patient (as shown in many other scRNAseq cancer studies), due to patient specific TME factors (e.g., somatic mutations). How are those patient-specific differences taken into account when looking for common patterns between cancer types?

→ We thank the reviewer for mentioning this issue. While our main focus in the manuscript was on non-epithelial cells, we understand the concern about how epithelial cells, including malignant ones, tend to cluster by patients in many scRNA-seq studies. As we mentioned in

response to comment #1, we applied NMF per sample to identify recurring gene expression programs. Recent pan-cancer single-cell studies (PMID: 37258682, 35931863) have shown that NMF is effective in revealing these patterns and helps in analyzing the inter-tumor heterogeneity within epithelial cell populations. To provide further clarification, we refer the reviewer to Figure S9A–9B (shown below). In this figure, we have visually represented various epithelial cell states and their respective organ of origins. Figure S9B clearly illustrates organ-specific clusters such as those enriched in the kidney. Simultaneously, some epithelial cell states, like those associated with cell cycling, are found in multiple organs. This supports that our analysis successfully identifies robust cell states present across different types of cancer. In summary, we believe our approach allows for a comprehensive exploration of common patterns that extend beyond specific cancer types.

[Distribution of epithelial cell states derived from diverse organs (Figures S9A–B)]

4. NMF method was extensively discussed in the literature, with pros and cons. When it comes to using NMF to find shared patterns between groups of cells, the field has been controversial in whether “the negative values were set to 0” is a proper preprocessing step

before applying NMF – it can almost be perceived as removing data to fit the assumption of NMF, which can be dangerous. Recognizing there are several big single-cell papers published in high-impact journals utilizing NMF for this purpose, it might not be reasonable to ask for re-analyzing data using an alternative method. However, authors may consider at least demonstrating that they can reproduce the same results using methods other than NMF – such as scCoGAPS (PMID: 31121116), which does not require removing data to fit the assumptions. scCoGAPS is only mentioned here as an example, other tools may serve this purpose, I will leave it to the authors' preference to address this question. Authors may consider demonstrating this in major cancer types if not all.

→ In response to the reviewer's suggestion, we applied scCoGAPS to various cell types of each patient. Due to computational constraints, we focused on breast, colon, liver, head and neck, lung, ovary, pancreas, and kidney datasets. To address the ongoing debate surrounding the practice of "setting negative values to 0" in NMF, we conducted our scCoGAPS analysis using a set of highly variable genes from a log-normalized gene-cell matrix. As demonstrated below, our scCoGAPS approach successfully identified robust cell states that exhibited significant correlations with cell states identified through NMF. Additionally, we conducted differential gene expression analysis with COSG (PMID: 35048116) to pinpoint specific gene expression patterns within cell types, further confirming the strong correlation between NMF-derived cell state scores and cell type-specific gene signatures (shown below).

[Correlations of scCoGAPS and COSG derived gene signatures with cell states identified in this study]

5. TLS has been previously reported to be associated with immunotherapy response. However, its prevalence is cancer-specific, for example, they are more prevalent in melanoma and lung, but less in bladder, or renal cell carcinoma, especially mature TLS. In addition to all datasets combined, have authors investigated in which cancer type(s) the immunotherapy response predictive power of TLS is detected (was it MEL, LC?)? How is this TLS signature different from known immunotherapy biomarkers, such as IFNG

signature, or are they confounded? Is TLS a surrogate of an IFNG-associated TME (e.g., highly correlated), or do these two signatures each act as independent biomarkers of immunotherapy response?

→ Please refer to the figure mentioned below (former Figure S14) for our analysis of the predictive power of cell states for each cancer type. In our LC cohort, the PD-L1 pathway exhibited the most favorable impact on immunotherapy, while in BLCA, MEL, and RCC, the CXCL9+ macrophage, antigen-presenting machinery, and CD160+ intraepithelial lymphocyte states, respectively, were associated with the most favorable responses to immunotherapy.

[Predictive power of cell states in immunotherapy cohort analyzed by each cancer type (former Figure S14)]

However, considering that well-known TLS components are more prominent in our LC cohort (as mentioned in comment #16 from Reviewer 1), we scrutinized all individual cohorts in terms of patient characteristics. As a result, it was brought to our attention that the number of responders in the individual cohorts examined in Figure 5D, except for our LC cohort, was limited (see the table below).

Cohort	Cancer type	Number of patients	Number of responders
David	Renal cell carcinoma	165	84
Gide	Melanoma	73	50

Hugo	Melanoma	26	14
Maria	Bladder cancer	347	104
Miao	Renal cell carcinoma	33	13
Riaz	Melanoma	46	21
SMC	Lung cancer	497	225
Van Allen	Melanoma	75	30

[Patient characteristics of the immunotherapy cohorts included in this work]

We suspected that this might have caused limited statistical power for the individual cancer types, that is, MEL, RCC, and BLCA, in our analysis of Figure S14. When an adequate number of responders is achieved by combining all cohorts, excluding our LC cohort, we obtain a similar pattern of cell states that significantly predict the response to immunotherapy, as depicted in the forest plot below.

Hence, while the prevalence of TLS varies among cancer types, we cannot definitively conclude that the influence of TLS varies among cancer types in our study due to the limited number of responders in the MEL, RCC, and BLCA cohorts treated with immunotherapy. However, considering the results of our meta-analysis, as shown below, it would be reasonable to anticipate that established TLS components could provide favorable responses to immunotherapy when a sufficient number of responders are gathered for each cancer type. This underscores the significance of our LC cohort, which includes the largest number of patient samples, and the need for pan-cancer analyses.

Due to the limited number of responders in the BLCA, MEL, and RCC cohorts, which may have introduced confounding results in Figure S14, we have decided to remove Figure S14 and its corresponding content from the manuscript.

[Predictive power of cell states in immunotherapy-treated cohorts]

Furthermore, in this pan-cancer analysis, which includes a sufficient number of responders, we discovered that both the IFN-gamma signature and well-known TLS components demonstrate favorable responses to immunotherapy. We have incorporated these findings into our revised manuscript and figures (Figure 5E).

[Predictive power of IFN-gamma signature in immunotherapy-treated cohorts (Figure 5E)]

Then, as the reviewer pointed out, we conducted a detailed examination of the association between TLS and IFN-gamma signatures. First, to identify TLS-enriched cell types and create a TLS signature (refer to comment #4 from Reviewer 3), we aimed to pinpoint cell types enriched to TLS using the RCC spatial transcriptome data and the information from pathologically defined TLS spots (PMID: 35231421). By comparing cell abundance between TLS and non-TLS spots, we characterized the cell types that are enriched in TLS, as

demonstrated below. Subsequently, by compiling the cell state genes from these TLS-enriched cell types, we established a TLS signature (available in Supplementary Table 7) that effectively distinguishes TLS from non-TLS spots.

[Defining TLS-enriched cell types & TLS signature (Figure 6B–6C)]

When we compared our TLS signature with the previously defined IFN-gamma signature (PMID: 31044165), we found that only a small proportion of genes overlapped between the two signatures (including *PLEK*, *UBD*, *FBLN1*, *JCHAIN*, *IRF8*, *KLF2*, *IDO1*, *PCDH9*, *CSF2RB*, *GBP2*, *NET1*, *HBG2*, and *NR3C1*).

[Intersection of TLS and IFN-gamma signatures]

Additionally, please take note that although we initially referred to the tumor ecosystem, consisting of interferon and tumor-infiltrating immune cells, including *LAMP3*⁺ DC, *CCL19*⁺ fibroblast, and Tfh, as the "TLS-associated community", we have found that other TLS-enriched cell types, such as *CD16*⁺/*XCL1*⁺ NK-cells, plasma cell, ILC3, and Th17, do not belong to this community (shown below). To prevent any further confusion, we have redefined this community as the "interferon-enriched community" since it encompasses multiple interferon-associated cell states or signatures derived from various cell types. We have updated our manuscript accordingly.

[Co-occurrence network in tumor tissues (Figure 5A)]

Regarding the association of the IFN-gamma signature with cell states incorporated into the interferon-enriched community (formerly the "TLS-associated community"), the IFN-gamma signature exhibited a significant correlation with interferon-related cell states derived from various cell types, as illustrated below (mesenchymal interferon, *IFIT1*⁺ interferon signaling, *ISG15*⁺ macrophage, and *Epi_interferon.Barkley*). Although TLS-enriched cell states, such as *LAMP3*⁺ DC, Tfh, and Treg, co-occurred with interferon states (Figure 5A, shown above), they did not highly correlate with the IFN-gamma signature as much as other interferon signatures, as depicted below. Additionally, TLS-enriched cell types that are not included in the interferon-associated community (Th17, ILC3, plasma cell, *PI16*⁺ fibroblast, and *CD16*⁺/*XCL1*⁺ NK cell), also did not show a high correlation with the IFN-gamma signature, as observed with other interferon signatures (shown below). Thus, despite both TLS-enriched cell types and interferon signatures showing favorable responses to immunotherapy, we conclude that it is not feasible to consider TLS as a surrogate for IFN-gamma, at least in our study.

[Correlation between IFN-gamma signature score and diverse cell states]

6. lastly, it will be important for the paper to provide raw data objects or files after authors' QC, but before applying integration, normalization, or batch correction methods. That way, users have the liberty to apply their own methods, which will broaden the impact of the data from this paper.

→ We thank the reviewer for encouraging us to share our analyzed datasets with the research community. As the reviewer requested, we provide post-QC raw data prior to any integration, normalization, or batch correction processes through the Figshare repository (DOI:10.6084/m9.figshare.24312331). Please kindly access through a private link for reviewers (<https://figshare.com/s/a51b42a7ab60b31aba2f>).

Reviewer #3 (Remarks to the Author): Expert in cancer genomics, scRNA-seq, tumour microenvironment, and immunotherapy

By integrating 30 cancer types with over 4.9 million unsorted single-cell transcriptomic data, J.E. Kang et al aimed to profile the tumor microenvironment at a higher resolution with large samples size. They further explored the signature molecular and cellular states, which may contribute to immunotherapy response outcome. The study has an impressive data integration and are organized carefully. However, the following points should be considered or explained.

→ We thank the reviewer for acknowledging the importance and the implications of this work. We have made efforts to address the points raised by the reviewer, and now feel that our paper has been improved considerably thanks to the constructive comments. We would also like to kindly inform that our study has been included as part of the Human Cell Atlas projects (HCA-96), and we have acknowledged this information in the Acknowledgement section.

1. The overall UMAP pattern is slightly different between the figure provided in manuscript and the website (<https://cellatlas.kaist.ac.kr/ecosystem/>). Some clusters seem to be manually deleted in the manuscript (see below). For instance, on the website, several epithelia clusters are located far from the others, especially one cluster is located on the top (see below), please explain why?

→ We apologize for any potential confusion arising from the discordant pattern observed between our manuscript and the website. The UMAP visualization on the website had not been updated to the latest version of our atlas. We have now updated the website to the latest version. Furthermore, we would like to inform that the outlier cluster which has been questioned by the reviewer primarily consists of tumor cells originating from the uvea with expression of well-known marker genes (PMID: 33577798, 31980621, shown below). For comprehensive information and visualization of our atlas, we kindly direct the reviewer to visit the updated website (<https://cellatlas.kaist.ac.kr/ecosystem/>).

[Visualization of cells according to organ-of origin & expression of uveal melanoma tumor marker genes]

2. Multiple subclusters were divided including a lot of novel annotated. Indeed, some of these “novel” clusters have been annotated with different names in previous reports, particularly functional annotated. For instance, P116+ fibroblast is more likely stem cell niche. Therefore, the “novel” clusters, which have been reported, should be carefully re-annotated.

→ As the reviewer pointed out, we acknowledge that several cell state annotations, which have been reported previously, could benefit from refinement. We searched for a myriad of literature to refine our annotations as listed below. As illustrated in the figure below, we identified significant correlations between our cell states and previously reported gene signatures. Subsequently, we made appropriate modifications to our manuscript and figures.

- GC-comitted B-cell → Lymphoblast (reference: <https://maayanlab.cloud/Enrichr/enrich?dataset=16e6c4cd276374dce5b16903f033def7>)
- Proliferating fibroblast → Cell cycling (PMID: 37258682)
- *CXCL10*⁺ fibroblast → Interferon response (PMID: 37258682)
- *COLEC11*⁺ fibroblast → Stellate (PMID: 30348985)
- *TNF*⁺ B-cell → Stress (PMID: 37258682)

[Correlation of cell states with previously reported signatures]

Regarding the *PI16*⁺ fibroblast, which was specifically questioned by the reviewer, we would like to clarify that *PI16*⁺ fibroblasts have been previously defined (PMID: 33981032), and they are not a novel cluster that we have newly annotated. It's important to note that the *PI16*⁺ fibroblasts defined in the previous study also exhibit a high correlation with our *PI16*⁺ fibroblast cell state, as shown above.

Additionally, concerning the *AKR1C1*⁺ and *WNT5A*⁺ inflammatory fibroblast populations, it's worth noting that although a previous study (PMID: 33622705) categorized *CXCL1–3*-expressing fibroblasts as a single inflammatory fibroblast population (pan-iCAFs-2), our analysis has revealed two distinct subpopulations that share common expressions of *CXCL1/3/8*, as illustrated below. However, these two populations significantly differed in terms of marker genes (*AKR1C1*, *FOSL1*, *LIF*, and *THAP2* vs. *WNT5A*, *GREM1*, *TNC*, and *MMP1*), tissue origins (normal vs. tumor), organ preferences (breast, ovary, and uterus vs. head and neck, and colon), cellular interactions, and spatial co-localization patterns. Consequently, we recognized the need for separate annotations for these two inflammatory

fibroblast populations, which we have designated as *AKR1C1*⁺ and *WNT5A*⁺ inflammatory fibroblasts, as *AKR1C1* and *WNT5A* account for the distinctions between them, as shown below.

[Dotplot of marker gene expression patterns between *AKR1C1*⁺ and *WNT5A*⁺ inflammatory fibroblasts (Figure 4A)]

[Marker gene expressions of *AKR1C1*⁺ and *WNT5A*⁺ inflammatory fibroblasts in UMAP (Figure S12C)]

3. The definition of inflammatory CAF in Figure 4 is confusion. For instance, P116+ fibroblast is more likely stem cell niche, which is highly enriched in normal/adjacent samples. Therefore, this cluster are more likely normal fibroblast rather than CAF, how can it be annotated as inflammatory CAF? To exclude such confusion, it is important to illustrate the proportion of normal/adjacent and tumor for each cluster.

→ We thank the reviewer for bringing this matter to our attention. After a comprehensive review of our manuscript, we have determined that a correction is indeed necessary. Given the differences in tissue enrichments observed among inflammatory fibroblast populations, as demonstrated below, we have replaced the term "inflammatory CAF" with "inflammatory fibroblasts" in relevant sections to avoid any potential confusion.

[Ro/e analysis showing tissue preferences of mesenchymal cell states (Figure 3E)]

4. TLS community is an interesting topic in this study. However, the information is very limited and confusing. Some issues should be considered or explained: 1. Most of the cell types included in Figure 5A-5C are not specific to TLS, how to distinguish the TLS located cells and whether some cell types are TLS-specific? 2. TLS is barely observed in normal tissue, but some cell types highlighted in this study (e.g., CXCL9+ macrophage) are still presented in normal, thus can't be claimed as TLS-specific. 3. It is better to construct a TLS signature and labeled in spatial transcriptome.

→ We thank the reviewer for raising such an essential issue that needs to be addressed. While we did not intend to imply that all cell types featured in Figure 5A–5C are exclusive to TLS (hence termed "TLS-associated" rather than "TLS-specific"), we fully concur with the reviewer on the importance of defining TLS-specific cell types. To address this concern, we leveraged the RCC spatial transcriptome, along with data from pathologically defined TLS spots (PMID: 35231421) as shown below.

[Examples of TLS-defined spots of the RCC spatial transcriptome data]

By comparing cell type abundances between TLS and non-TLS spots, we identified cell types significantly enriched in TLS. Subsequently, by compiling the cell state genes from these TLS-enriched cell types, we established a TLS signature (available in Supplementary Table 7) that effectively distinguishes TLS from non-TLS spots.

[Defining TLS-enriched cell types & TLS signature (Figure 6B–6C)]

Using our TLS signature, we applied it to multiple spatial transcriptome datasets as demonstrated below, and identified significant co-localization with TLS-enriched cell types across various cancer types.

[Spatial distribution of TLS-enriched cell types and TLS signature (Figure 6D)]

Additionally, please take note that although we initially referred to the tumor ecosystem, consisting of interferon and tumor-infiltrating immune cells, including *LAMP3*⁺ DC, *CCL19*⁺ fibroblast, and Tfh, as the "TLS-associated community", we have found that other TLS-enriched cell types, such as *CD16*⁺/*XCL1*⁺ NK-cells, plasma cell, ILC3, and Th17, do not belong to this community (shown below). To prevent any further confusion, we have redefined this community as the "interferon-enriched community" since it encompasses multiple interferon-associated cell states or signatures derived from various cell types. We have updated our manuscript accordingly.

[Co-occurrence network in tumor tissues (Figure 5A)]

5. according to Fig. 6C, ISG15+ macrophage ranks the highest correlation with composite signature, but is lacking in the spatial demonstration in Fig. 6B.

→ In addition to comment #4, we have now developed a TLS signature instead of the previous composite signature. For detailed information, please consult the newly defined TLS signature and refer to Figure 6C–6D (shown above in comment #4).

6. this study claimed several novel cell subclusters, some of which are small in population according to the proportion analysis. IHC or multiplexed immunofluorescence (mIF) is needed to establish its real existence. In addition, some cell subtypes (e.g., CXCL9+ macrophage) were considered to be located in TLS. Since TLS is much easier to be revealed through IHC/mIF rather than spatial transcriptome, it is better to illustrate the localization of these cells in TLS structure through experimental approach with several examples.

→ For experimental validation, we conducted RNA single-molecule fluorescence *in situ* hybridization (smFISH) targeting *WNT5A*, *GREM1*, and *PDGFRA* to detect *WNT5A*⁺ inflammatory fibroblasts in tissue samples from colorectal cancer (CRC) and head and neck cancer (HNSC). We confirmed the presence of *WNT5A*⁺ inflammatory fibroblasts in CRC and HNSC tissue samples through the overlapping signal from the *WNT5A*, *GREM1*, and *PDGFRA* RNA probes in desmoplastic stromas, as illustrated in Figure 4E and S13A (attached below). We also aimed to validate the existence of interferon-producing mesenchymal cells, as mesenchymal interferon cell states demonstrated favorable responses to immunotherapy. RNA smFISH targeting *CXCL10* and *PDGFRA* effectively

identified the presence of interferon-producing mesenchymal cells in HNSC tissue sample. We have updated the experimental findings in our manuscript and figures accordingly. Furthermore, in response to comment #4, we believe that the determination of TLS-enriched cell types using the RCC spatial transcriptome (PMID: 35231421) addresses the concern raised by the reviewer.

[*In situ* RNA sm-FISH targeting *WNT5A*, *PDGFRA*, and *GREM1* (Figure 4E)]

[*In situ* RNA sm-FISH targeting *WNT5A*, *PDGFRA*, and *GREM1* (Figure S13A)]

[*In situ* RNA sm-FISH targeting *CXCL10* and *PDGFRA* (Figure S15)]

7. the repository dataset in Zenodo is restricted. Although “Reviewers are allowed to access our repository” but when I request the access, name and email address should be provided. In addition, GSE218989 can’t be found in GEO database, and the followup information of the lung cancer cohort is missing, which may be provided as supplementary table.

→ We apologize for any inconvenience related to accessing Zenodo. To address this issue, we have uploaded our analyzed datasets and codes to Figshare (DOI:10.6084/m9.figshare.24312331) and are providing a Private link:

Private link: <https://figshare.com/s/a51b42a7ab60b31aba2f>

Furthermore, please note that GSE218989 is currently set to private status and will be made accessible upon the publication of our paper. To enable reviewers to access the data, we are sharing the link and access token as follows:

Link: <https://www.ncbi.nlm.nih.gov/geo/query/acc.cgi?acc=GSE218989>

Token: whetgieapnshvgp

Additionally, we have included updated overall survival and progression-free survival information for our lung cancer cohort as a supplementary table. Kindly refer to the attached Supplementary Table 8 for details (shown below).

PatientID	Responder	Overall survival (days)	Death	Progression-free survival (days)
SMC__Pat1	1	1177	0	312
SMC__Pat10	1	1163	0	1163
SMC__Pat100	0	285	1	128

SMC__Pat101	0	38	1	34
SMC__Pat102	1	839	0	249
SMC__Pat103	0	162	1	57
SMC__Pat104	1	0	0	472
SMC__Pat105	0	129	1	69
SMC__Pat106	0	336	1	91
SMC__Pat107	0	0	0	55
SMC__Pat108	0	0	1	57
SMC__Pat109	1	0	0	620
SMC__Pat11	0	62	1	21
...				

[Response, survival patient characteristics of SMC cohort (Table S8)]

8. Line 842, the algorithm is missing.

→ We apologize for the missing component in our manuscript. The algorithm did not appear due to an error in the process of converting the Word file to a PDF document. We now have revised the manuscript as shown below:

Previous line 842: A chi-square-derived expected number was obtained, and Ro/e was calculated as $\log_2 \frac{\text{observed}}{\text{expected}}$. We offer the calculation method in the screenshot figure displayed below, as a precaution in case the conversion doesn't function again.

$$\log_2 \frac{\text{observed}}{\text{expected}}$$

REVIEWER COMMENTS

Reviewer #2 (Remarks to the Author):

The authors have thoroughly addressed my comments. The NMF analysis within each patient is clearly described and addressed the batch bias issue across multiple cohorts, and the new analysis demonstrating TLS cell states do not correlate with IFNG signature was also particularly important.

Two last comments I had were:

1 – for the new Figure 5E, it was a bit unclear what those cell states (or signatures) mean on the y-axis. Can authors provide a more elevated description on those terms/categories? (this is just an example – what does “Mesenchymal interferon” mean?)

2 - it is great that authors have deposited their data objects and analysis codes. Would be better if authors could also provide documentation describing those files, for example, what is “br_spatial_GSE210616_GSM6433585.h5ad.xz”. This is obvious to scientists with computational background but perhaps less so for the general audience. It would also be helpful for authors to provide a step-by-step documentation describing the analysis steps using codes from the “analysis_code” folder in order to reproduce the analysis, input and output files, etc.

Reviewer #3 (Remarks to the Author):

The revised manuscript has greatly improved. Several issues are listed below.

1. Introducing AND-gating algorithm through citing the related reference or provide the method package/web link.
2. TLS and the related TLS signature has been highlighted in this study, which is of great importance. However, TLS was not identified in all tumor types according to previous reports, it will be necessary to illustrate to TLS signature separately in different cancer types with scRNA-seq or bulk seq data, and investigate whether the TLS signature score is higher in TLS enriched cancer type, thus support the accuracy of TLS signature constructed in this study.
3. As TLS has been considered as a favorite prognostic marker for cancer treatment, it will be interesting to deconvolute the TLS signature with the bulk transcriptome data of the immunotherapy cohort included in this study to estimate its impact.
4. Provide the gene list of Figure 2A.

5. Cite a reference to support SPP1 is “M1/M2 polarization-related”.
6. Something strange for the myofibroblast analysis, a lot of literatures have indicated the enrichment of myofibroblast in tumor (e.g., PMID: 32393771, 22143274, 29455927, 36171287, 37059066, 36333338) which is opposite to the results in this study. it’s necessary to provide evidence (such as some published studies) to support the evenly distribution of myofibroblast in non-malignant and malignant tissue, rather than just stating “in contrast to the previous reports”. In addition, discussion or explanation of this difference should be provided.
7. Which cancer type was used to analysis the cell-cell communications in Figure 4D? Given the opposite direction AKR1C1 expression in different cancer types illustrated in Figure 4C, interactions analysis should also be performed separately (at least provided as supplementary figures) in terms of cancer type because the interaction analysis is mainly rely on gene expression.
8. The result of smFISH, which was used to indicate real existence of WNT5A+ fibroblast and its co-localization with macrophage, is frustrating. Overlapping signal is barely observed. Zoom in figure with arrow labeling should be provided the overlapped signal.

EDITORIAL NOTE

Reviewer #3 considered that the following original concerns from Reviewer #1 were not fully addressed in your revision:

6. Please further clarify how the malignancy state was defined; particularly if the CNV-score was used.
7. The Reviewer did not raise specific concerns, but your reply could be further explained.
9. Please provide the correct supplementary table numbers in the file and in your manuscript.
10. Is the result consistent with the previous paired comparison?
11. Please explain why epithelial and neural cells were combined to perform the subclustering analysis.
12. Figure labels were hard to read, the response could be further simplified.
14. Brain vascular endothelial cells were still present in unexpected cancer types; please provide an explanation.

REVIEWER COMMENTS

Reviewer #2 (Remarks to the Author):

The authors have thoroughly addressed my comments. The NMF analysis within each patient is clearly described and addressed the batch bias issue across multiple cohorts, and the new analysis demonstrating TLS cell states do not correlate with IFNG signature was also particularly important.

Two last comments I had were:

1 – for the new Figure 5E, it was a bit unclear what those cell states (or signatures) mean on the y-axis. Can authors provide a more elevated description on those terms/categories? (this is just an example – what does “Mesenchymal interferon” mean?)

→ We appreciate the reviewer for providing valuable comments to enhance the interpretability of our results. The y-axis labels represent the cell states identified in our study or previously defined signatures (please refer to Table S2) and only those statistically associated with the response to immunotherapy are visualized. The x-axis depicts the odds ratio of the investigated cell state and signatures. To address the reviewer's concern, we have updated the figure legend below:

“Forest plot of immunotherapy-response predictive cell states and gene signatures from other studies through meta-analysis of immunotherapy-treated bulk RNA-seq cohorts. The x-axis represents the odds ratio, in which the dotted vertical line represents an odds ratio of 1, and the y-axis denotes cell states and previously defined gene signatures deconvoluted in the meta-analysis. Only cell states that achieved statistical significance are depicted. Cell states with odds ratios greater than 1 are those associated with favorable responses to immunotherapy. The colors correspond to the cell type categories of each cell state. APM, antigen-presenting machinery; DC, dendritic cell; EMT, epithelial-to-mesenchymal transition; IFNG, interferon-gamma; NK, natural killer cell; pDC, plasmacytoid dendritic cell; Treg, regulatory T cell.”

As for the “mesenchymal interferon” state, which was specifically mentioned by the reviewer, this state refers to the mesenchymal-derived interferon state in Table S2. We acknowledge that the term “mesenchymal interferon” might have confused readers. Consequently, we have modified it to “interferon (mesenchymal)” to mitigate any potential confusion.

2 - it is great that authors have deposited their data objects and analysis codes. Would be better if authors could also provide documentation describing those files, for example, what is “br_spatial_GSE210616_GSM6433585.h5ad.xz”. This is obvious to scientists with computational background but perhaps less so for the general audience. It would also be helpful for authors to provide a step-by-step documentation describing the analysis steps using codes from the “analysis_code” folder in order to reproduce the analysis, input and output files, etc.

→ We agree with the reviewer’s comment that a detailed description of our files and codes would be more helpful to the general audience. To address this, we have included descriptions of deposited files and comments in analysis codes for the benefit of research community as exemplified below.

Dataset	Organ	Accession	No. of samples
bc_spatial_GSE171351	Bladder Cancer	GSE171351	4
br_spatial_GSE210616	Breast Cancer	GSE210616	43
br_spatial_GSE198745	Breast Cancer	GSE198745	1
br_spatial_ZEN4739739	Breast Cancer	ZEN4739739	6
crc_spatial_crlm	Colo-Rectal Cancer	http://www.cancerdiversity.asia/scCRLM/	8
hn_spatial_GSE181300	Head and Neck Cancer	GSE181300	8
lu_spatial_GSE189487	Lung Cancer	GSE189487	2
lu_spatial_GSE200916	Lung Cancer	GSE200916	6
ov_spatial_GSE211956	Ovarian Cancer	GSE211956	8
ov_spatial_GSE213699	Ovarian Cancer	GSE213699	8
pan_spatial_GSE203612	Pan Cancer	GSE203612	8
rcc_spatial_GSE175540	Renal Cell Carcinoma	GSE175540	24
sscc_spatial_GSE144239	Skin Cancer	GSE144239	4
sscc_spatial_GSE221390	Skin Cancer	GSE221390	7

Sample names are assigned in the format 'Dataset name'_'sample accession ID'.h5ad.xz (e.g., br_spatial_GSE210616_GSM6433585.h5ad.xz)
Detailed information for each sample can be accessed through the GEO accession or website link

[Documentation for spatial datasets]

```

#Identifying top 50 genes for each NMF module cluster
marker={}
for f in set(adata.obs['leiden']):
    temp=adata.raw.to_adata()
    sub=temp[temp.obs['leiden']==f]
    marker['leiden_{}'.format(f)]=list(pd.DataFrame(np.sum(sub.X,axis=0),index=temp.var_names).sort_values(by=0,asce
df = pd.DataFrame.from_dict(marker,orient='index').T

```

```

#Identifying module clusters enriched mitochondrial or ribosomal genes
slim_df=df[:50]
mito_ribo=[]
for f in slim_df.columns:
    if len([x for x in slim_df[f] if x.startswith('MT-')])>=2:
        mito_ribo.append(f.split('_')[-1])
    elif len([x for x in slim_df[f] if x.startswith('RP')])>=2:
        mito_ribo.append(f.split('_')[-1])

```

```

mitoriborm=[]
for f in adata.obs['leiden']:
    if f in mito_ribo:
        mitoriborm.append('Mito_Ribo')
    else:
        mitoriborm.append('NMF_Module')
adata.obs['mito_ribo']=mitoriborm

```

```

#Identifying module clusters derived from single dataset
single=[]
for f in set(adata.obs['leiden']):
    temp=adata[adata.obs['leiden']==f]
    if len(set(temp.obs['Dataset']))==1:
        single.append(f)

```

```

singdata=[]
for f in adata.obs['leiden']:
    if f in single:
        singdata.append('Single_dataset')
    else:
        singdata.append('NMF_Module')
adata.obs['single_data']=singdata

```

```

scjpc.us(adata,'single_data')

```

```

#Identifying ambient RNA/soup effect clusters
rmlist=[]
for f in sorted(set(adata.obs['leiden'])):
    print('leiden_{}'.format(f))
    temp=adata[adata.obs['leiden']==f]
    abundant_organ=sorted(Counter(temp.obs['Organ_orig']).items(),key = lambda x:-x[1])

```

[Analysis codes with step-by-step documentation]

```

#Identifying mean gene weights for each cell type cluster
sdata = adata
anno_key = 'Celltype'
mks = scjp.markers.marker(sdata,anno_key)

cts = sorted(set(sdata.uns['cdm_'+anno_key]['mean']))

vstack = []
for ct in cts:
    vstack.append(sdata.uns['cdm_'+anno_key]['mean'][ct])

means = np.vstack(vstack)

#Ligand-receptor gene selection
col_ord = ['CD44:SIGLEC15','IL6:IL6R','TNFRSF12A:TNFSF12','IL24:IL20RA','WNT5A:FZD5','WNT5A:PTK7',
           'WNT5A:ROR2','GREM1:ACVR1','GREM1:ACVRL1']

cols = ['AKR1C1+ inflammatory fibroblast:CTSK+ macrophage (osteoclast)',
        'AKR1C1+ inflammatory fibroblast:DC1',
        'AKR1C1+ inflammatory fibroblast:PRR-induced mo-DC',
        'AKR1C1+ inflammatory fibroblast:ILC3',
        'WNT5A+ inflammatory fibroblast:Cancer cell',
        'WNT5A+ inflammatory fibroblast:Desmoplastic fibroblast',
        'WNT5A+ inflammatory fibroblast:WNT5A+ inflammatory fibroblast',
        'WNT5A+ inflammatory fibroblast:BMP4+ intestinal fibroblast',
        'WNT5A+ inflammatory fibroblast:Endothelial']
][::-1]

#Visualization of ligand-receptor interaction analysis
from sklearn.preprocessing import MaxAbsScaler
final = np.zeros(shape=(len(col_ord),len(cols)))
for i, x in enumerate(col_ord):
    l = x.split(':')[0]
    r = x.split(':')[1]
    left_idx = sdata.raw.var_names==l
    right_idx = sdata.raw.var_names==r
    j = -1

```

[Analysis codes with step-by-step documentation]

```

#Steps
#1. Import data and response information.
#2. Execute logistic regression independently for each cohort with respect to CCA modules.
#3. Aggregate coefficients across cohorts through meta-analysis for each CCA module.
#4. Generate a forest plot illustrating the results of CCA modules.

#Import CCA module score calculated by ssgsea
data = read.table('Immunotherapy_cohort_CCA_module_231123_total/gseapy.samples.normalized.es.txt',
stringsAsFactors = F, header = T, sep = '\t', check.names = F)

#Import response information data
response = read.table('Sample_Efficacy.txt', stringsAsFactors = F, header = T, sep = '\t')
colnames(response) = c('PatientID', 'Efficacy')

#Annotate cohort information
response$Cohort = sapply(response$PatientID, function(x){unlist(strsplit(x, split = '__'))[1]})

#Annotate cancer type information
response$Type <- case_when(
  response$Cohort %in% c('SMC') ~ 'Lung',
  response$Cohort %in% c('Van_Allen', 'Hugo', 'Riaz', 'Gide') ~ 'Melanoma',
  response$Cohort %in% c('David', 'Miao') ~ 'RCC',
  response$Cohort %in% c('Maria') ~ 'Bladder',
  TRUE ~ 'NA')

#Remove NA samples
response = response[-which(response$Type == 'NA'),]

#Statistics template table
stat_table = data.frame(t(c('Term', 'OR', 'OR.low', 'OR.upper', 'OR.pval')))
colnames(stat_table) = c('Term', 'OR', 'OR.low', 'OR.upper', 'OR.pval')

```

[Analysis codes with step-by-step documentation]

We have updated our analysis codes and datasets in the Zenodo repository (DOI:10.5281/zenodo.10260049). Please kindly access through a private link for reviewers (<https://zenodo.org/records/10260049?token=eyJhbGciOiJIUzUxMiIsImIhdCI6MTcwMTgzNzA3NSwiZXhwIjoxNzA2NjU5MTk5fQ.eyJpZCI6IjM0NDZlZGZkLTgyNTgtNDdlZi1hMjliLTU2YjI4NDFiZjI4NSIsImRhdGEiOnt9LCJyYW5kb20iOiI3Y2I4OTQyMWE5Mjk2ODY0YThlZWl2MWM3ZTM2OWVjYyJ9.IMOoZZHT031PmJg3uWnhXjwIZRKpOwWvNDVDaRoZpW92WodUn7OkjfDSnTXQyuXGIC-qNXUdxzRtSoXv-i3jDQ>).

Reviewer #3 (Remarks to the Author):

The revised manuscript has greatly improved. Several issues are listed below.

1. Introducing AND-gating algorithm through citing the related reference or provide the method package/web link.

→As pointed out by the reviewer, we have cited relevant references to the AND-gating algorithm (PMID: 37332931) in the methods section.

2. TLS and the related TLS signature has been highlighted in this study, which is of great importance. However, TLS was not identified in all tumor types according to previous reports, it will be necessary to illustrate to TLS signature separately in different cancer types with scRNA-seq or bulk seq data, and investigate whether the TLS signature score is higher in TLS enriched cancer type, thus support the accuracy of TLS signature constructed in this study.

→We thank the referee for highlighting this crucial point. Given that the majority of cell types illustrated in Figure 6B are "TLS-enriched" rather than "TLS-specific," we postulated that merely assembling the cell state genes from TLS-enriched cell types might not accurately reflect TLS biology and activity.

[TLS-enriched cell types in RCC spatial transcriptome (Figure 6B)]

Hence, to pinpoint genes that are specifically associated with TLS, we conducted differential expression analysis between TLS and non-TLS spots (illustrated below) with PyDESeq2 (PMID: 37669147) and derived a TLS signature consisting of genes that are highly “TLS-specific”.

[Examples of TLS-defined spots of the RCC spatial transcriptome data]

[Volcano plot of differentially expressed genes between TLS and non-TLS spots]

To test our TLS signature in various cancer types, as suggested by the reviewer, we applied our TLS signature to multiple cancer types in the TCGA cohort. We found that Diffuse Large B-cell Lymphoma (Lymphoid Neoplasm) demonstrated the highest TLS signature, suggesting a strong correlation with lymph node structures. Subsequently, lung cancer exhibited elevated TLS signatures, while glioblastoma multiforme, adrenocortical carcinoma, and brain lower grade glioma displayed low TLS signature scores.

Furthermore, we conducted a comparison between our TLS signature and previously reported TLS signatures (PMID: 31942071 and 31092904) as shown below, revealing comparable results with our TLS signature.

[Fridman et al., TLS signature (PMID: 31092904)]

[Cabrita et al., TLS signature (PMID: 31942071)]

3. As TLS has been considered as a favorite prognostic marker for cancer treatment, it will be interesting to deconvolute the TLS signature with the bulk transcriptome data of the immunotherapy cohort included in this study to estimate its impact.

→ We thank the reviewer for bringing this matter to our attention. Consistent with previous findings indicating TLS as a positive prognostic marker for immune checkpoint blockade, our TLS signature also predicted favorable responses to immunotherapy, as depicted in the updated manuscript and figures below:

“Our TLS signature also predicted favorable responses to immunotherapy in immunotherapy-treated cohorts (p-value < 0.001; Figures 6C and S19A).”

[Predictive power of TLS signature in immunotherapy-treated bulk RNA-seq cohorts
(Figure 6C)]

[Predictive power of TLS signature in immunotherapy-treated cohorts (Figure S19A)]

4. Provide the gene list of Figure 2A.

→ The full gene lists of Figure 2A are provided in Table S4. Some screenshots of Table S4 are attached for your reference.

Gene	Breast CD8+ T-cell Foldchange	Breast CD8+ T-cell adjusted p-value	Colon CD8+ T-cell Foldchange	Colon CD8+ T-cell adjusted p-value	Head and Neck CD8+ T-cell Foldchange	Head and Neck CD8+ T-cell adjusted p-value
TNFRSF18	1.12268417	0.0296399633	0.8993910088	3.23E-68	1.348550393	3.74E-108
UTS2	2.633805729	0.06234567897	2.078657937	1.55E-153	1.949125729	1.05E-99
TNFRSF9	1.977808785	0.0003667605181	0.9639840638	3.79E-56	2.277228851	1.38E-200
ZNF683	3.300025919	1.77E-15	1.524330023	7.45E-111	1.507770328	7.05E-148
SNHG3	0	1	0	1	-4.70599694	0
HDAC1	0.7942917886	0.02899815856	-0.1437110155	1	0.7148905324	5.08E-82
CLSPN	1.599685384	0.1341969882	1.152230791	1.29E-29	1.353867924	5.95E-43
CDC48	1.143387335	0.5313767692	0.6830230462	7.41E-09	1.196076804	2.45E-26
KIF2C	1.811198156	0.2823944742	0.8003405511	5.09E-11	1.338216732	5.59E-28
CDKN2C	1.483307685	0.1263260121	1.351111377	2.90E-66	1.239035996	2.31E-42
LINC01358	2.025530235	0.2320516407	0.8189057631	1.93E-10	1.512245263	1.13E-26
DNAB1B4	1.27359634	0.007699471341	0.7664229423	3.36E-39	-0.07113374916	1
IFI44L	3.412452544	1.14E-11	1.388577146	1.76E-59	3.722629996	0
IFI44	2.399123963	2.36E-06	1.064149726	1.33E-49	2.041106866	5.59E-267
GBP3	1.286866881	0.09930178261	0.27804524	1	0.6071584935	1.06E-24
GBP1	2.383918119	2.27E-08	-0.1109015758	1	1.095866507	1.36E-180
GBP2	1.29554658	3.01E-06	0.3490059041	1	0.5015783444	2.29E-63
GBP4	1.523019517	0.001778204701	0.2980746369	1	1.392891778	1.69E-156
GBP5	4.345437151	5.19E-16	0.06650492451	1	1.413935439	1.14E-306
CSF1	1.87315019	0.00641543871	1.193535438	3.92E-43	1.607039126	2.75E-113
CHI3L2	1.142250854	0.6255320616	-0.7029520271	4.51E-26	1.847806728	1.59E-129
ANKRD35	2.65997862	0.03985916758	0.7883450475	2.79E-15	1.590145305	1.32E-47

[Universal hallmark genes of CD8⁺ T cells elevated in tumor tissues]

Gene	Breast Fibroblast Foldchange	Breast Fibroblast adjusted p-value	Colon Fibroblast Foldchange	Colon Fibroblast adjusted p-value	Head and Neck Fibroblast Foldchange	Head and Neck Fibroblast adjusted p-value
SAMD11	1.534410187	4.03E-102	1.053229345	1.00E-219	0.09628600819	1
HES4	-0.006896798315	1	0.6833136561	2.77E-143	1.500707098	1.22E-67
ISG15	1.734991166	0	1.52544119	0	2.765324767	2.95E-175
MXRA8	2.301526185	0	0.8835985637	0	0.07264495944	1
ANKRD65	2.436285181	0	0.6337136222	2.42E-38	-1.125529469	4.63E-20
FNDC10	1.698672883	3.13E-103	1.328111024	6.64E-126	1.094646097	0.002968343204
MMP23B	1.381952095	4.13E-290	0.67735438	1.13E-151	0.4162793063	1
MEGF6	1.794040425	1.97E-208	0.609120108	1.25E-30	1.179066457	2.68E-07
ICMT	2.191476193	0	0.8713475483	5.44E-144	0.2945618002	1
PLOD1	3.041027001	0	0.1611054051	1	1.675013664	8.19E-38
VPS13D	0.7053911257	2.31E-53	0.4838059286	1	-1.001209002	1.15E-13
PDPN	-0.771780238	7.43E-278	2.002173914	0	2.33790222	7.65E-88
FBLIM1	2.166134825	0	0.3061042948	1	1.029996139	4.72E-12
ARHGEF19	1.236896267	9.75E-80	0.5657381343	1.86E-25	1.348862754	3.21E-08
MFAP2	4.385071837	0	-0.8169800871	0	2.257854425	1.80E-101
CAPZB	1.048838692	0	0.5440799337	0	0.5103302636	2.81E-30
NBL1	0.8217386442	0	1.531931013	0	0.7719250199	7.09E-51
PLA2G2A	1.661425959	6.82E-112	0.7015165046	3.78E-61	-1.19974904	2.53E-36
PLA2G5	1.71827459	1.83E-109	1.111118256	2.23E-79	-2.307899651	6.78E-80
HSPG2	1.854916918	0	1.47515862	0	-1.078448322	1.14E-99
MAN1C1	1.277469572	1.36E-105	0.9850882004	6.27E-100	-2.170797662	3.57E-93
PTPRU	0.9868918626	1.08E-59	0.7890106491	6.52E-40	0.3296948713	1
COL16A1	2.075435664	0	0.9605565812	0	0.6935479374	4.45E-11
SPOCD1	2.408158842	1.35E-198	2.672950686	0	2.488724636	5.92E-10
AZIN2	2.572683079	0	0.9732814833	1.31E-76	0.9410068502	0.0001613854247
CSMD2	1.903042697	3.44E-113	1.234400952	6.35E-93	1.084182831	0.005337388419

[Universal hallmark genes of fibroblasts elevated in tumor tissues]

5. Cite a reference to support *SPP1* is "M1/M2 polarization-related".

→ We conducted a comprehensive review of relevant literature and identified a previous study outlining the role of *SPP1*⁺ macrophage in both M1/M2 polarization (PMID: 37535729). However, more references indicated the association of *SPP1*⁺ macrophages with M2 signatures (PMID: 33545035, 28830685), and revised our manuscript as follows:

"In particular, tumor-infiltrated macrophages universally expressed immune checkpoint (IL4I1), M2 polarization-related (*SPP1*; ref PMID: 33545035, 28830685), and inflammatory genes (*CCL7*, *ADAMDEC1*, and *SLAMF9*), whereas dendritic cells in tumors were found to exhibit elevated expression of *CCL19* and *LAMP3*, which are associated with inflammatory and migratory functions (Figures 2A–2B)."

6. Something strange for the myofibroblast analysis, a lot of literatures have indicated the enrichment of myofibroblast in tumor (e.g., PMID: 32393771, 22143274, 29455927, 36171287, 37059066, 36333338) which is opposite to the results in this study. it's necessary to provide evidence (such as some published studies) to support the evenly distribution of myofibroblast in non-malignant and malignant tissue, rather than just stating "in contrast to the previous reports". In addition, discussion or explanation of this difference should be provided.

→ We annotated myofibroblasts based on the prior pan-cancer fibroblast study (PMID: 33622705; *ACTA2*, *MYH11*, *MYLK*, *MCAM*, and *TAGLN*). The term "myCAF" was used in this reference as it classified fibroblasts only in cancer tissues. While many studies mentioned by the reviewer discussed the immunosuppressive role of myofibroblasts in cancer, their role in cancer progression, and therapy resistance, they did not assess the relative enrichment of myofibroblasts in tumor tissues compared with normal tissues.

We conducted a more in-depth analysis of a study (PMID: 36333338), which includes both tumor and normal tissues. In this paper, c1, c2, and c4 were collectively referred to as "CAFmyo". Among these, c1 (*TAGLN*, *ACTA2*, *MYL9*, *MYH11*, *MYLK*) corresponds to the myofibroblasts annotated in our study, and c2 (*COL1A1*, *MMP11*, *CTHRC1*, *COL12A1*, and *POSTN*) was annotated as desmoplastic fibroblasts (based on PMID: 33622705). C4 (*DCN*, *FBLN1*, *DPT*, *MGP*) was annotated as *P116*⁺ fibroblasts (based on PMID: 33981032) as depicted below. Thus, our annotation further dissected the heterogenous myofibroblast-like CAF population and defined myofibroblast with specific marker sets. Following the reviewer's suggestion, we assessed the tumor-normal enrichment patterns of the myofibroblast defined in our study and their consistency with previous reports as shown below:

[NMF weight of marker genes of myofibroblast, desmoplastic fibroblast, and *P116*⁺ fibroblast in mesenchymal cell state UMAP]

[Mesenchymal cell state UMAP]

When our results were compared with the aforementioned previous study (PMID:36333338), the tumor-normal enrichment of the myofibroblasts (c1 cluster)

appears comparable, as shown in the attached figure below. Additionally, the tumor-normal enrichment patterns for c2 and c4 are similar to our study.

[Tumor-normal enrichment of cell type clusters in the study PMID:36333338 (Figure 1E)]

It is important to note that our study encompassed both tumor and normal samples from various organs, and the tissue enrichments of myofibroblasts in Figure 3E represent a comprehensive outcome across all organs. Tumor-normal enrichment of myofibroblasts in breast, pancreas, liver, lung, and kidney exhibits enrichment in tumors compared with normal tissue as shown below. This may account for the previous literature (PMID: 32393771, 29455927, 36171287) outlining the tumorigenic role of myofibroblasts in these organs. However, in the colon and stomach, myofibroblasts were similarly found between tumor and normal tissues, aligning with prior studies that have documented the presence of myofibroblasts in the normal intestine (PMID: 21252048, 27165847). The tumor-normal enrichment of myofibroblasts in each organ is also provided in Figure S17C and we also included an extra paragraph in the discussion section as written below:

[Tumor-normal enrichment of myofibroblasts across organs]

“Moreover, myofibroblasts in our study exhibited relatively higher enrichment in normal tissues compared with the findings of the previous pan-cancer study (PMID: 36333338). It is worth noting that our study encompassed both tumor and normal samples from various organs, and the tissue enrichments of myofibroblasts represent a comprehensive outcome across all organs (Figure 3E). Organ-wise analysis revealed myofibroblast enrichment in tumor tissues of the breast, pancreas, liver, lung, and kidney (Figure S17C), suggesting a potential tumorigenic role of these fibroblasts in these organs (PMID: 32393771, 29455927, 36171287). However, tumor-normal enrichment of myofibroblasts in the colon and stomach was comparable, aligning with prior studies that have documented the presence of myofibroblasts in the normal intestine (PMID: 21252048, 27165847).”

7. Which cancer type was used to analysis the cell-cell communications in Figure 4D?

Given the opposite direction AKR1C1 expression in different cancer types illustrated in

Figure 4C, interactions analysis should also be performed separately (at least provided as supplementary figures) in terms of cancer type because the interaction analysis is mainly rely on gene expression.

➜We appreciate the reviewer's insightful observation. Ligand-receptor interaction analysis in Figure 4D was initially performed in breast, ovary, colon, and head and neck tumor tissue samples. We have updated our manuscript as follows:

"To gain insight into how these inflammatory fibroblasts shape the TME, we performed ligand-receptor analysis to unravel distinct interactions of *AKR1C1*⁺ and *WNT5A*⁺ inflammatory fibroblasts with other cell types in BRCA, CRC, HNSC, and OV samples (Figures 4D and S12D–S12E)."

Additionally, we conducted ligand-receptor analysis for each cancer type as depicted below. We observed comparable cell-cell interaction patterns between cancer types (BRCA & OV, and CRC & HNSC).

[Ligand-receptor analysis of *AKR1C1*⁺ inflammatory fibroblast in BRCA and OV (Figure S12D)]

[Ligand-receptor analysis of *WNT5A*⁺inflammatory fibroblast in CRC and HNSC (Figure S12E)]

8. The result of smFISH, which was used to indicate real existence of *WNT5A*⁺ fibroblast and its co-localization with macrophage, is frustrating. Overlapping signal is barely observed. Zoom in figure with arrow labeling should be provided the overlapped signal.

→ We appreciate this valuable feedback. Regarding the RNA smFISH results, we identified the presence of *WNT5A* expressing fibroblasts with overlapping signals of *PDGFRA*, *WNT5A*, and *GREM1* but did not intend to identify its co-localization with macrophages. In response to the reviewer's suggestion, we have updated our RNA smFISH figures with a zoom-in figure, including arrow labeling, as shown below.

[*In situ* RNA sm-FISH targeting *WNT5A*, *PDGFRA*, and *GREM1* (Figure 4E)]

[*In situ* RNA sm-FISH targeting *WNT5A*, *PDGFRA*, and *GREM1* (Figure S13A)]

Comments regarding Reviewer #1

6. Please further clarify how the malignancy state was defined; particularly if the CNV-score was used.

→ We have updated how malignant cells were identified in detail in the methods section as written below.

Copy number variation inference for identification of malignant cells

Single-cell transcriptome-based large-scale copy number variation (CNV) of malignant cells was inferred using inferCNVpy (using the package available at <https://github.com/icbi-lab/infercnvpy>) with default window size and gencode v29 for genomic location reference. We employed `infercnvpy.tl.infercnv` to infer CNV and selected normal immune cells or fibroblasts as reference normal cells depending on each cancer type. Following dimensional reduction (`infercnvpy.tl.pca`) and clustering based on CNV profiles (`infercnvpy.tl.leiden`), cells were visualized on a CNV UMAP (`infercnvpy.tl.umap`) and CNV scores were calculated using `infercnvpy.tl.cnv_score`. Putative malignant cells were defined based on two criteria: i) Formation of separate clusters, a known property of malignant cells (PMID: 27124452), and ii) higher CNV scores compared with known normal cell types (normal epithelial, fibroblasts, or immune cells based on each cancer type).

7. The Reviewer did not raise specific concerns, but your reply could be further explained.

→ The original concern raised in comment #7 from Reviewer #1 was related to the inclusion of rare cells or states for the deconvolution of bulk transcriptome analysis (e.g., deconvoluting brain vascular endothelial in lung cancer, as mentioned in comment #14 below) that that might not be necessarily important or meaningful due to their low abundance.

To prevent the deconvolution of cell states representing rare populations, we established the Ro/e filtering threshold for each cell state (Ro/e filtering threshold =

mean(Ro/e) – 1 s.d(Ro/e)). If the tumor-derived Ro/e values for specific cell states within an organ did not surpass the filtering threshold, we identified those cell states as rare and not meaningful within that organ. This approach was employed to exclude the deconvolution of rare cell states that are not relevant to the survival analyses of specific organs (e.g., alveolar fibroblasts in colon cancer & glioblastoma).

9. Please provide the correct supplementary table numbers in the file and in your manuscript.

→The supplementary table asked by the reviewer (organ-specific CAF gene signatures), was omitted during the initial revision since re-analysis of survival data with a median split revealed that these signatures lacked prognostic significance (refer to comment 8 from Reviewer #1). We also thoroughly examined the supplementary table links in our manuscript and identified that all links were appropriately associated with relevant supplementary tables.

10. Is the result consistent with the previous paired comparison?

→We have summarized the number of differentially expressed tumor-upregulated genes compared with adjacent normal or healthy normal tissues as shown in the table below.

	Tumor vs adjacent normal	Tumor vs healthy normal	Intersect
CD8⁺ T cell	221	303	139
CD4⁺ T cell	24	23	10
Treg	17	23	13
Dendritic cell	51	59	16
Endothelial	339	554	254
Fibroblast	524	847	323

Macrophage	354	329	198
NK cell	26	12	9

[Number of genes upregulated in tumor when compared to adjacent normal and healthy normal tissues]

Overall, we observed a considerable overlap among differentially expressed genes in diverse cell types. Considering the changes in conditions for the differential expression analysis (both in tissues; adjacent normal vs. normal, and in sample size), we posit that variations in the results are acceptable. However, given that well-known immune checkpoints (e.g., *PDCD1*, *TIGIT*, *CTLA4*, and *CXCL13*) and cancer-associated fibroblast genes (e.g., *COL1A1*, *FAP*, *MMP11*, *CTHRC1*) exhibit consistent results in both differential expression analyses, it suggests that tumor-specific gene expression programs have been well identified in our results.

[Tumor vs. adjacent normal (previous Figure 2B)]

[Tumor vs. normal (current Figure 2B)]

11. Please explain why epithelial and neural cells were combined to perform the subclustering analysis.

→ We thank the reviewer for mentioning this issue. It is worth mentioning that not only in our study but also in recent pan-cancer single-cell studies that utilized NMF analysis (PMID: 37258682, 35931863), neural cells were combined. NMF analysis decomposes a gene-cell matrix into recurring gene expression programs, allowing for the identification of factors (or NMF modules) in a gene-cell matrix even when epithelial and neural cells are combined. Additionally, considering that both epithelial and neural tissues originate and differentiate from the ectoderm, we propose that it is plausible to combine both cell types when performing sub-clustering analysis.

12. Figure labels were hard to read, the response could be further simplified.

→ We apologize for the inconvenience. In response to the reviewer's comment, we have included high-resolution figures to enhance interpretability. Initially, the reviewer raised concerns about the unusually high abundance of small populations (e.g., neutrophils) in spatial transcriptome datasets.

To address this, we calculated cell proportions in scRNA-seq datasets and cell abundance in spatial transcriptomes across cancer types as depicted below, and observed comparable results between the two measures. Notably, we identified significant "inter-tumoral" heterogeneity, a well-known feature of tumors (PMID: 29115304 and 23299535). This finding explains the high abundance of neutrophils in specific spatial transcriptome samples and the variations in the cell proportions between the single-cell atlas and spatial transcriptome.

[Comparison of cell proportion in BRCA scRNA-seq and abundance proportion in BRCA spatial transcriptome]

[Comparison of cell prortion in OV scRNA-seq and abundance proportion in OV spatial transcriptome]

[Comparison of cell proportion in SSCC scRNA-seq and abundance proportion in SSCC spatial transcriptome]

[Comparison of cell proportion in RCC scRNA-seq and abundance proportion in RCC spatial transcriptome]

14. Brain vascular endothelial cells were still present in unexpected cancer types; please provide an explanation.

→ To address this important point, we thoroughly examined our cell states and identified that brain vascular endothelial cell states were derived from several metastasis samples (e.g.; lung cancer brain metastasis sample, melanoma brain metastasis sample). To mitigate this issue, we performed organ of origin Ro/e analysis based on the primary tumor, adjacent normal, and healthy normal samples, and excluded NMF modules derived from metastasis samples. We have revised the figure accordingly, as shown below.

[Ro/e analysis showing tissue/organ preferences of mesenchymal cell states (Figure S17C)]

Overall survival

[Prognostic impact of mesenchymal states in relevant TCGA cancer types (Figure S17D)]

REVIEWERS' COMMENTS

Reviewer #2 (Remarks to the Author):

The authors have addressed both of my comments. The expanded description of figure legend as well as the sharing of analysis codes and more file documentation are very helpful.

Reviewer #3 (Remarks to the Author):

All comments have been well addressed except the smFISH figure. Also the arrows were added, but the overlapped signal is still barely seen. The signals looks like mutually exclusive. Zoom in images with 20x or 40x should be provided

REVIEWER COMMENTS

Reviewer #3 (Remarks to the Author)

All comments have been well addressed except the smFISH figure. Also the arrows were added, but the overlapped signal is still barely seen. The signals looks like mutually exclusive. Zoom in images with 20x or 40x should be provided

→ We thank the reviewer for this valuable comment. We have identified issues where the fluorescence signal seemed to weaken during the conversion of figure images to PDF and have taken measures to address this.

[*In situ* RNA sm-FISH targeting *WNT5A*, *PDGFRA*, and *GREM1* (original Figure 4E)]

[*In situ* RNA sm-FISH targeting *WNT5A*, *PDGFRA*, and *GREM1* (original Figure S13A)]

To illustrate the overlap of signals from *WNT5A*, *PDGFRA*, and *GREM1*, we present zoom-in images of the RNA sm-FISH in areas indicated by arrows and identified a clear co-occurrence of *WNT5A*, *PDGFRA*, and *GREM1* signals, as shown below.

Figure 4E, top

Figure 4E, bottom

[Zoom-in images of *In situ* RNA sm-FISH targeting *WNT5A*, *PDGFRA*, and *GREM1* (areas f or Figure 4E)]

Figure S13A, top

Figure S13A, middle

Figure S13A, bottom

[Zoom-in images of *In situ* RNA sm-FISH targeting *WNT5A*, *PDGFRA*, and *GREM1* (areas of or Figure S13A)]

Furthermore, we explored other regions for the co-occurrence of signals from *WNT5A*, *PDGFRA*, and *GREM1*, and identified additional spots, as illustrated below.

[*In situ* RNA sm-FISH targeting *WNT5A*, *PDGFRA*, and *GREM1*]

Collectively, we have observed a clear overlap of *WNT5A*, *PDGFRA*, and *GREM1* signals across multiple regions and samples. However, we acknowledge the reviewer's point about potential non-overlapping signals in the original figures, due to low resolution, and we have updated our RNA smFISH figures with the images displayed above. We have ensured that high resolution is maintained during the PDF conversion.